# GeoSplatting: Towards Geometry Guided Gaussian Splatting for Physically-based Inverse Rendering

## Abstract

We consider the problem of physically-based inverse rendering using 3D Gaussian Splatting (3DGS) representations (Kerbl et al., 2023b). While recent 3DGS methods have achieved remarkable results in novel view synthesis (NVS), accurately capturing high-fidelity geometry, physically interpretable materials, and lighting remains challenging, as it requires precise geometry modeling to provide accurate surface normals, along with physically-based rendering (PBR) techniques to ensure correct material and lighting disentanglement. Previous 3DGS methods resort to approximating surface normals but often struggle with noisy local geometry, leading to inaccurate normal estimation and suboptimal material-lighting decomposition. In this paper, we introduce GeoSplatting, a novel hybrid representation that augments 3DGS with explicit geometric guidance and differentiable PBR equations. Specifically, we bridge isosurface and 3DGS together, where we first extract isosurface mesh from a scalar field, then convert it into 3DGS points and formulate PBR equations for them in a fully differentiable manner. In GeoSplatting, 3DGS is grounded on mesh geometry, enabling precise surface normal modeling, which facilitates the use of PBR frameworks for material decomposition, achieving excellent decomposition performance, especially for reflective cases. This approach further maintains the efficiency and quality of NVS from 3DGS while ensuring accurate geometry from the isosurface. Comprehensive evaluations across diverse datasets demonstrate the superior efficiency and competitive inverse rendering performance of GeoSplatting compared to state-of-the-art inverse rendering baselines.

## 1 Introduction

The inverse rendering task, *i.e.*, recovering 3D attributes such as geometry, spatially-varying materials, and lighting from multi-view images or videos, has been a long-standing goal in computer vision and graphics. It plays a critical role in numerous industrial applications, including film production, gaming, and VR/AR, for photo-realistic novel-view synthesis and immersive user interactions. This task is typically approached using carefully designed 3D representations (Mildenhall et al., 2020; Müller et al., 2022; Wang et al., 2021; Shen et al., 2021; 2023) coupled with the corresponding differentiable rendering techniques (Boss et al., 2021a; Verbin et al., 2022a; Chen et al., 2019; Laine et al., 2020). While great progress has been made recently (Munkberg et al., 2022; Jiang et al., 2023; Gao et al., 2023), efficiently and accurately capturing various 3D attributes remains challenging due to the complexities of light transport in real-world environments, including intricate local geometry, non-Lambertian surface, complex lighting conditions, occlusions, *etc*.

The key to tackling the inverse rendering task lies in effectively modeling the underlying 3D geometry, where Physically-Based Rendering (PBR) techniques can be applied to disentangle materials and lighting. Numerous prior works have developed various 3D representations and their corresponding differentiable rendering equations to address this challenge, each offering unique advantages and limitations. Implicit representations like NeRF (Mildenhall et al., 2020; Verbin et al., 2022b) are well-suited for novel view synthesis but are computationally expensive and incompatible with existing graphics pipelines. In contrast, explicit representations like mesh (Munkberg et al., 2022) provide explicit geometry, allowing for well-defined rendering techniques, and facilitating tasks like relighting and material editing. However, optimizing explicit representations is challenging,

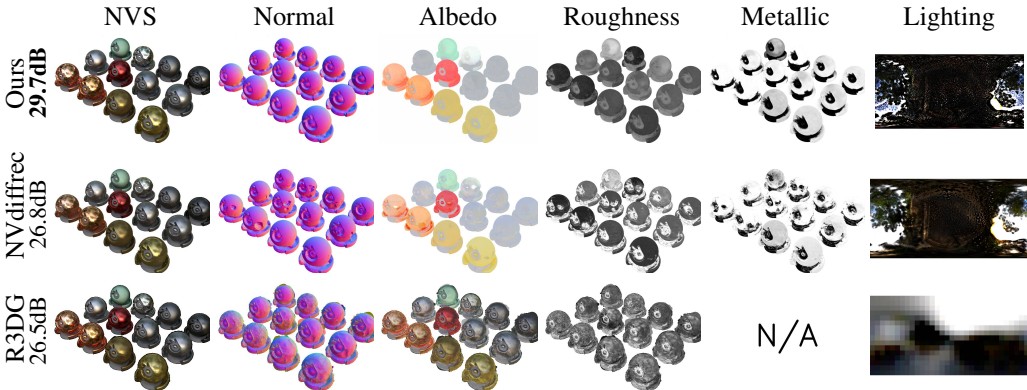

Figure 1: We propose GeoSplatting, a novel inverse rendering approach that augments Gaussian Splatting with explicit geometric guidance. GeoSplatting enables more accurate geometry recovery, achieving excellent material and lighting decomposition performance and superior efficiency. Please zoom in for details.

especially when dealing with complex geometries like thin structures. More recently, 3D Gaussian Splatting (3DGS) (Kerbl et al., 2023b) has emerged as an efficient 3D representation for high-quality novel-view synthesis. However, vanilla 3DGS is not designed to provide accurate geometry or disentangled materials, limiting its applicability to inverse rendering tasks. To tackle this challenge, various methods have studied assigning a normal direction to each 3DGS point to model local geometric surfaces, along with a PBR formula using the approximated normals (Jiang et al., 2023; Shi et al., 2023; Gao et al., 2023). However, these approaches offer only approximations of true normals, and as a result, may struggle with local minima in regions of complex geometry.

In this paper, we propose GeoSplatting, a more principled solution that leverages the strengths of both explicit representations and 3DGS for the inverse rendering task. At the core of GeoSplatting is a differentiable adaptor that integrates differentiable isosurface techniques (Shen et al., 2021; 2023) with 3D Gaussian Splatting. Specifically, we first utilize the differentiable isosurface techniques to extract a mesh from a scalar field that we want to optimize. We then introduce MGadapter, *i.e.*, Mesh-to-Gaussian-adaptor, that samples 3D Gaussian points on the mesh surface in a differentiable manner, naturally grounding the location of each Gaussian point on the surface geometry, from which we could estimate precise normal for each point. To render the sampled 3D Gaussian points, we design an efficient and differentiable PBR framework, leveraging the split-sum model (Karis, 2013) and applying it to the 3D Gaussian points. During training, since all the operations are differentiable, we can train our model end-to-end.

Our GeoSplatting offers several advantages over both 3DGS and explicit mesh-based representations. Compared to the vanilla 3DGS (Kerbl et al., 2023b) and its variants, our approach provides explicit geometric guidance from the isosurface, enabling more accurate normal estimation, which is crucial for inverse rendering optimization. On the other hand, compared to the mesh representations (Munkberg et al., 2022), GeoSplatting leverages the high efficiency and superior rendering quality inherited from 3DGS. Moreover, while the concept of constraining 3DGS with geometry is not new (Yu et al., 2024; Xiang et al., 2024), existing methods typically rely on a discrete optimization strategy where the implicit SDF field (Wang et al., 2021) and Gaussian points are learned separately. In contrast, GeoSplatting explicitly guides Gaussian points with the isosurface and can be optimized in an end-to-end fashion, reducing training time and improving inverse rendering quality.

In summary, our key contributions are as follows:

1. We introduce a novel inverse rendering framework, GeoSplatting, which bridges 3DGS with isosurface techniques, achieving competitive decomposition performance with state-of-the-art methods.

2. Leveraging geometry guidance, GeoSplatting delivers significantly superior rendering quality and achieves the highest normal accuracy on reflective surfaces compared to all inverse rendering baselines.

3. GeoSplatting significantly outperforms all inverse rendering baselines in training time, showcasing its excellent efficiency in capturing material-lighting interactions.

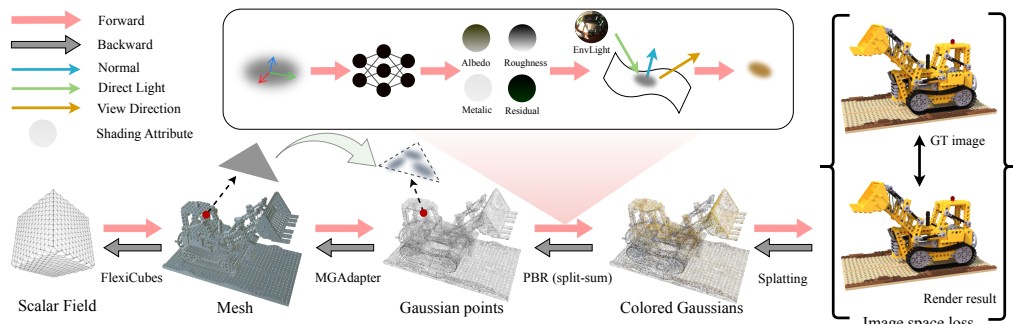

Figure 2: **Pipeline**. GeoSplatting first extracts an intermediate mesh from the scalar field, upon which Gaussian points are sampled and rendered using PBR equations. Finally, they are composited into images through the Gaussian rasterization pipeline. The entire process is fully differentiable.

## 2 RELATED WORK

**NeRF-based Inverse Rendering** Since Neural Radiance Field (NeRF) (Mildenhall et al., 2020) represented scenes using implicit neural networks and leveraged differentiable volume rendering (Drebin et al., 1988) to achieve highly detailed novel view synthesis, a large amount of subsequent work has extended it to inverse rendering tasks, by enhancing both the representations and the rendering equations to introduce more physical constraints, allowing the separation of physically-based rendering attributes such as normals, materials, and lighting (Boss et al., 2021a;b; Yariv et al., 2021; Wang et al., 2021; Zhang et al., 2021a; 2022; Verbin et al., 2022a; Liang et al., 2023; Jin et al., 2023; Ge et al., 2023). While NeRF-based methods encode BRDF in a single radiance field, the use of implicit representations and volume rendering makes them difficult to integrate into existing graphics pipelines, as well as requires dense sampling that leads to slow rendering speed.

**Mesh-based Inverse Rendering** To address the rendering efficiency issue, explicit representations (*e.g.*, mesh (Shen et al., 2021; 2023)) combined with differentiable rendering techniques (Chen et al., 2019; 2021; Laine et al., 2020) have demonstrated their ability to extract explicit geometry, material, and lighting from multi-view images (Munkberg et al., 2022; Hasselgren et al., 2022). However, mesh optimization is typically more challenging than implicit fields, which often struggle with complex geometry (*e.g.*, thin structures), especially under high-frequency lighting-material interactions (*e.g.*, specular objects).

**3DGS-based Inverse Rendering** Recently, 3D Gaussian Splatting (3DGS) (Kerbl et al., 2023a) has emerged as a powerful representation for novel view synthesis with significantly superior rendering efficiency to NeRF and its variants. While it can be promising to utilize 3DGS to build more powerful inverse rendering framework, its geometry is often misaligned with the ground truth surface and prone to floaters, leading to inability in capturing accurate lighting-material interactions. To improve this, a branch of research extends the vanilla 3DGS with various geometry enhancement techniques by either introducing additional regularizations to ensure Gaussian points closely adhere to the surface and keep them squeezed like disks (Guédon & Lepetit, 2024; Huang et al., 2024; Dai et al., 2024), or jointly training 3DGS with implicit SDF fields (Lyu et al., 2024; Yu et al., 2024; Xiang et al., 2024). Inspired by these geometry enhancement techniques, there are many methods extend the vanilla 3DGS for inverse rendering tasks. For instance, R3DG (Gao et al., 2023) learns additional normal attributes and regularizes normal directions using rendered depth maps. GS-Shader (Jiang et al., 2023) utilizes the shortest axis direction, while GIR (Shi et al., 2023) employs eigen decomposition to determine surface orientations. While most of these methods produce reasonable normals by conducting alpha-blending on the per-Gaussian normal attributes, such a blending process encounter much difficulty in obtaining precise normal, which prevent the methods from capturing high-frequency lighting-material interactions, leading to failure cases when modeling reflective surfaces. In contrast, we combine explicit mesh representation and 3DGS in a unified framework and train the framework in an end-to-end, similar to concurrent work Lin & Li (2024), but employ a novel mesh-to-Gaussian adaptor to achieve geometric alignment, enabling effective material and lighting decomposition, resulting in significantly better quality of both normal estimation and reflective surface modeling, therefore bringing excellent inverse rendering performance.

## 3 METHODOLOGY

We now present a detailed description of our method. In Sec. 3.1, we introduce geometry-guided Gaussian Splatting, where the Gaussian points are generated on the isosurface. Next, in Sec. 3.2, we extend the standard Gaussian rendering equations by incorporating physically-based rendering (PBR) to account for higher-order lighting effects. Finally, in Sec. 3.3, we discuss the training strategies, loss functions, and other key implementation details.

### 3.1 GEOMETRY GUIDED GAUSSIAN POINTS GENERATION

**Background**  In the vanilla 3DGS (Kerbl et al., 2023b) paper, a Gaussian ellipsoid is represented by a full 3D covariance matrix $\mathbf{\Sigma}$ and its center position $\boldsymbol{\mu}$: $G(\mathbf{x}) = e^{-\frac{1}{2}(\mathbf{x}-\boldsymbol{\mu})^T \mathbf{\Sigma}^{-1}(\mathbf{x}-\boldsymbol{\mu})}$, where $\mathbf{x}$ is the location of a 3D point. To ensure a valid positive semi-definite covariance matrix, $\mathbf{\Sigma}$ is decomposed into the scaling matrix $\mathbf{S}$ and the rotation matrix $\mathbf{R}$ that characterizes the geometry of a 3D Gaussian. Beyond $\boldsymbol{\mu}$, $\mathbf{S}$ and $\mathbf{R}$, each Gaussian maintains additional learnable parameters including opacity $o \in (0, 1)$ and Spherical Harmonic (SH) coefficients in $\mathbb{R}^k$ representing view-dependent colors ($k$ is related to SH order). During optimization, 3DGS adaptively controls the Gaussian distribution by splitting and cloning Gaussians in regions with large view-space positional gradients, as well as the culling of Gaussians that are nearly transparent. However, each Gaussian point is independent of others and lacks global geometry constraints, which often leads inaccurate surfaces and floaters.

**Method**  Our goal is to introduce explicit geometric guidance to 3D Gaussian Splatting. To this end, we propose GeoSplatting, which leverages isosurface techniques (Shen et al., 2021; 2023) to constrain Gaussian points to the mesh surface. Specifically, we hope to optimize a scalar function $\zeta : \mathbb{R}^3 \to \mathbb{R}$, which may be discretized directly as values at grid vertices or evaluated from an underlying neural network, *etc*. We employ FlexiCubes (Shen et al., 2023) as the underlying geometric representation, allowing for the extraction of an intermediate triangle mesh $\mathbf{M}$ from $\zeta$ in a differentiable manner.

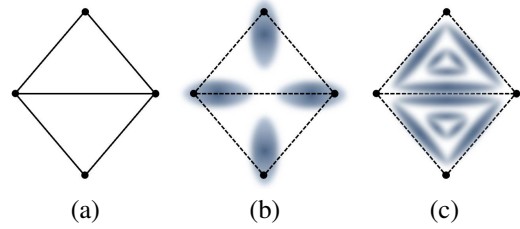

**Figure 3: MGadapter Overview**. Given surface triangles (a), we initially place one Gaussian point at each vertex (b), then densely draw six Gaussian points on each face (c).

With the intermediate mesh $\mathbf{M}$ as the explicit guidance, we then propose **MGadapter** $\mathcal{T}$ that samples Gaussian points from surface triangles in a differentiable manner to provide a bridge between mesh and Gaussian points. Specifically, the MGadapter $\mathcal{T}$ constrains the opacity and shape of the Gaussian points to ensure their alignment with the mesh $\mathbf{M}$.

We first determine the location of 3D Gaussian points from the isosurface mesh $\mathbf{M}$. As illustrated in Fig. 3, we explored various strategies and finally chose an adaptive way. At the start of the optimization, since the scalar function $\zeta$ is randomly initialized and the initial mesh contains numerous small faces, we only assign one Gaussian point to each vertex to reduce memory usage and accelerate training. As the shape quickly begins to converge, we switch to a face-based strategy, where we empirically place six Gaussian points on each triangle face in order to capture high-frequency geometric and texture details.

The opacity for each Gaussian point is set to one, and the position $\boldsymbol{\mu}$, scale $\mathbf{S}$, and rotation $\mathbf{R}$, along with the normal $\mathbf{n}$, are determined by the local geometry (vertices or faces):

$$(\boldsymbol{\mu}, \mathbf{S}, \mathbf{R}, \mathbf{n}) = \mathcal{T}(\mathbf{M}). \tag{1}$$

Specifically, the position $\boldsymbol{\mu}$ is a barycentric interpolation of mesh vertices, the normal $\mathbf{n}$ equals the normal of corresponding mesh faces, the scale $\mathbf{S}$ and the $\mathbf{R}$ are heuristic functions relative to $\boldsymbol{\mu}$, which ensure that the shortest axis of the Gaussian point aligns with $n$ and maintain a size that adequately covers the triangle.

Additionally, since the boundaries of Gaussian ellipsoids extend beyond their center points $\boldsymbol{\mu}$, we allow $\boldsymbol{\mu}$ to move slightly along the

GT Image          GT Normal

3DGS Normal   Our Normal

Figure 4: **Geometry-guided Normal Estimation**.

normal direction $\mathbf{n}$ to better align with the surface. This surface adjustment $v$, which is crucial for novel view synthesis, can be automatically learned via hash grids (Müller et al., 2022). Further details of MGadapter can be found in Appendix B.

**Discussion** The geometry-guided MGadapter offers significant advantages over both 3DGS (Kerbl et al., 2023a) and mesh-based representations (Munkberg et al., 2022) . First, compared to 3DGS, the geometry guidance from the isosurface provides more accurate geometry and precise surface normals without any depth or normal regularization terms, as shown in Fig. 4. This, in turn, significantly enhances the performance of inverse rendering tasks compared to prior works that rely on approximated normals. Moreover, transitioning from a mesh representation to Gaussian Splatting, *i.e.*, apply Gaussain-based rendering rather than mesh-based rendering, allows us to leverage the efficiency and representational capacity of Gaussian Splatting, enabling GeoSplatting to achieve much faster optimization time and superior novel view synthesis performance compared to NVdiffrec (Munkberg et al., 2022) as shown in our experiments.

### 3.2 PHYSICALLY-BASED GAUSSIAN RENDERING

The vanilla 3DGS assigns each Gaussian point a $k$-order spherical harmonic parameter to represent basic color and view-dependent rendering effects. In our GeoSplatting, we hope to represent high-order lighting effects with PBR materials.

**Background** We utilize the physically-based rendering equation (Kajiya, 1986) and GGX microfacet model (Walter et al., 2007) as follows:

$$L_o(\boldsymbol{\omega}_o) = \int_{\mathcal{H}^2} f_r(\boldsymbol{\omega}_i, \boldsymbol{\omega}_o) L_i(\boldsymbol{\omega}_i) |\mathbf{n} \cdot \boldsymbol{\omega}_i|, \mathrm{d}\boldsymbol{\omega}_i \tag{2}$$

$$f_r(\boldsymbol{\omega}_i, \boldsymbol{\omega}_o) = \frac{\mathbf{a}}{\pi} + \frac{D(\boldsymbol{\omega}_i, \boldsymbol{\omega}_o) F(\boldsymbol{\omega}_i, \boldsymbol{\omega}_o) G(\boldsymbol{\omega}_i, \boldsymbol{\omega}_o)}{4|\mathbf{n} \cdot \boldsymbol{\omega}_i||\mathbf{n} \cdot \boldsymbol{\omega}_o|} \tag{3}$$

In Eq. 2, the outgoing radiance $L_o(\boldsymbol{\omega}_o)$ in the direction $\boldsymbol{\omega}_o$ is computed as the integral of the BRDF function $f_r(\boldsymbol{\omega}_i, \boldsymbol{\omega}_o)$, the incoming light $L_i(\boldsymbol{\omega}_i)$, and the cosine term $|\mathbf{n} \cdot \boldsymbol{\omega}_i|$, which accounts for the angle between the surface normal $\mathbf{n}$ and the incoming light direction $\boldsymbol{\omega}_i$, over the hemisphere $\mathcal{H}^2$. In Eq. 3, the GGX model defines the BRDF function $f_r(\boldsymbol{\omega}_i, \boldsymbol{\omega}_o)$ as two components: the diffuse term $\frac{\mathbf{a}}{\pi}$ and the specular term $\frac{D(\boldsymbol{\omega}_i, \boldsymbol{\omega}_o) F(\boldsymbol{\omega}_i, \boldsymbol{\omega}_o) G(\boldsymbol{\omega}_i, \boldsymbol{\omega}_o)}{4|\mathbf{n} \cdot \boldsymbol{\omega}_i||\mathbf{n} \cdot \boldsymbol{\omega}_o|}$. The specular term models view-dependent specular surface reflection using the normal distribution function (NDF) $D$, the Fresnel term $F$, and the geometric attenuation $G$. To evaluate Eq. 2 efficiently, approximation methods such as splitsum (Karis, 2013) or spherical Gaussian (Chen et al., 2021) are often used to bypass the need for the extensive Monte Carlo sampling process. Following prior works (Munkberg et al., 2022; Shi et al., 2023), we use a split-sum model:

$$L_o(\boldsymbol{\omega}_o) \approx \int_{\mathcal{H}^2} f_r(\boldsymbol{\omega}_i, \boldsymbol{\omega}_o) |\mathbf{n} \cdot \boldsymbol{\omega}_i| \, \mathrm{d}\boldsymbol{\omega}_i \int_{\mathcal{H}^2} L_i(\boldsymbol{\omega}_i) D(\boldsymbol{\omega}_i, \boldsymbol{\omega}_o) |\mathbf{n} \cdot \boldsymbol{\omega}_i| \, \mathrm{d}\boldsymbol{\omega}_i \tag{4}$$

Eq. 4 enables fast pre-computation. The left BRDF term can be stored in a 2D lookup table and queried using $|\mathbf{n} \cdot \boldsymbol{\omega}_i|$ and $r$, while the right term is represented by pre-integrated environment maps and can also be queried by $r$. This allows directly computing outgoing radiance $L_o$ by the material parameters without any ray sampling, significantly improving rendering speed. For more details of PBR materials, please refer to (Karis, 2013; Munkberg et al., 2022).

**Method** Our goal is to add PBR materials to Gaussian points to produce high-order rendering effects, while still leveraging the efficient Gaussian rasterization pipeline. To achieve this, we compute the color of each Gaussian point using PBR equations, followed by Gaussian rasterization that renders the final image through alpha compositing.

Specifically, we assign each Gaussian point three PBR material parameters: a diffuse color $k_d = \frac{\mathbf{a}}{\pi} \in \mathbf{R}^3$, a roughness scalar $r$, and a metallic factor $m$. The roughness $r$ determines the GGX normal distribution function (NDF) $D$, while the metallic factor $m$ controls the specular highlight color $k_s$ by interpolating between plastic and metallic appearances: $k_s = (1 - m) \times 0.04 + m \times k_d$. Finally, the color $c_i$ of the $i$-th Gaussian point is computed as:

$$c_i = c_i^d + c_i^s + c_i^r \tag{5}$$

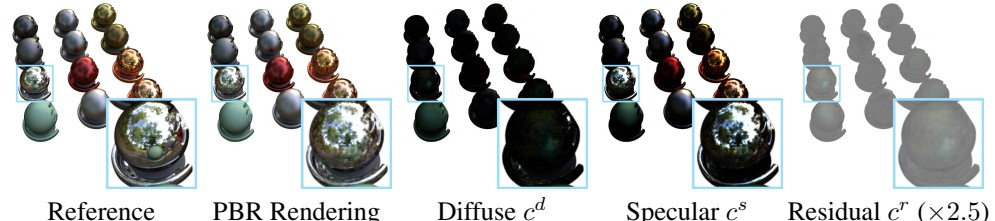

| Reference | PBR Rendering | Diffuse $c^d$ | Specular $c^s$ | Residual $c^r$ (×2.5) |

Figure 5: **PBR Rendering Decomposition**. Our PBR framework successfully disentangles the materials and lighting, capturing meaningful diffuse, specular and residual terms (Eq. 5). Note the residual image even learns the inter-reflection effects (the most shiny ball reflects a small green ball).

where $c_i^d$ and $c_i^s$ represent the diffuse and specular components computed using the split-sum model. In addition, we learn a residual color $c_i^r$ (Jiang et al., 2023) to account for high-order indirect lighting effects. Once the color computation is complete, we apply the efficient Gaussian rasterization pipeline to render the points into an RGB image $I$ and an alpha map $M$:

$$I = \sum_{i=1}^{N} c_i \alpha_i \prod_{j=1}^{i-1}(1 - \alpha_j), \quad M = \sum_{i=1}^{N} \alpha_i \prod_{j=1}^{i-1}(1 - \alpha_j) \tag{6}$$

where $\alpha$ is the projected opacity of each Gaussian. During rendering, we cull Gaussian points on back faces by checking the angle between surface normals and view directions.

**Discussion** Our PBR pipeline enables both high-order PBR effects and efficient rendering speed. First, the PBR model allows us to capture specular, view-dependent lighting effects. Additionally, the residual component helps model inter-reflection effects beyond the capabilities of the split-sum model. Moreover, since the color is computed at each Gaussian point, the fast Gaussian rasterization pipeline can be directly employed to generate the final images. Fig. 5 shows examples of the decomposition of each lighting component of our methods.

### 3.3 IMPLEMENTATION DETAILS

**Modeling PBR Attributes** GeoSplatting constrains the shape and opacity of each Gaussian point based on the corresponding surface triangle, and learns the PBR material attributes by querying MLP (Müller et al., 2022). Specifically, for the geometry, we optimize a $96^3$ grid using Flexi-Cubes (Shen et al., 2023). For each Gaussian point, we query its corresponding surface movement $v$, diffuse color $k_d$, roughness $r$, specular $k_s$ and residual color $c^r$ from a spatial MLP (Müller et al., 2022) $F$: $(v, k_d, r, m, c^r) = F(\boldsymbol{\mu})$. Additionally, we learn a $6 \times 512 \times 512 \times 3$ environment map to model the lighting. Details of the network architecture are provided in Appendix B.5.

**Loss Functions** GeoSplatting is a fully differentiable pipeline that can be trained end-to-end. Thanks to the geometry guidance, we do not require any surface or normal regularization terms, such as dist loss (Barron et al., 2022) or pseudo depth normal loss (Jiang et al., 2023; Gao et al., 2023), and the network can be supervised by photometric loss. However, similar to NVdiffrec, GeoSplatting also relies on an object mask loss, as optimizing the surface is more challenging. Specifically, the photometric loss is computed as: $\mathcal{L}_{photo} = \mathcal{L}_1 + \lambda_{ssim}\mathcal{L}_{SSIM} + \lambda_{mask}\mathcal{L}_{mask}$, where $\mathcal{L}_{mask} = \mathcal{L}_2(M_{pred}, M_{gt})$. Here we set $\lambda_{ssim} = 0.2$ and $\lambda_{mask} = 5.0$. Furthermore, we add an entropy loss to constrain the shape, following DMTet and FlexiCubes (Shen et al., 2021; 2023). To achieve better decomposition, we apply light regularization and smoothness regularization on $k_d$ and $k_s$, following NVdiffrec and R3DG (Munkberg et al., 2022; Gao et al., 2023). The final loss $\mathcal{L}$ is a combination of the photometric loss and the regularization losses: $\mathcal{L} = \mathcal{L}_{photo} + \lambda_{entropy}\mathcal{L}_{entropy} + \lambda_{smooth}\mathcal{L}_{smooth} + \lambda_{light}\mathcal{L}_{light}$. Details are provided in Appendix. C.

**Second Stage Optimization** GeoSplatting optimizes the underlying SDF while producing Gaussian points for image rendering. However, due to grid resolution limitations, we find that mesh-produced Gaussian points still struggle to capture detailed textures and thin geometric structures. To address this, we introduce a second-stage optimization where the Gaussian points are freely optimized without being constrained by the mesh. In the first stage, the mesh-generated Gaussian points already provide a refined shape and materials. Building on this, the second-stage optimization further enhances geometry and texture details, achieving better decomposition performance.

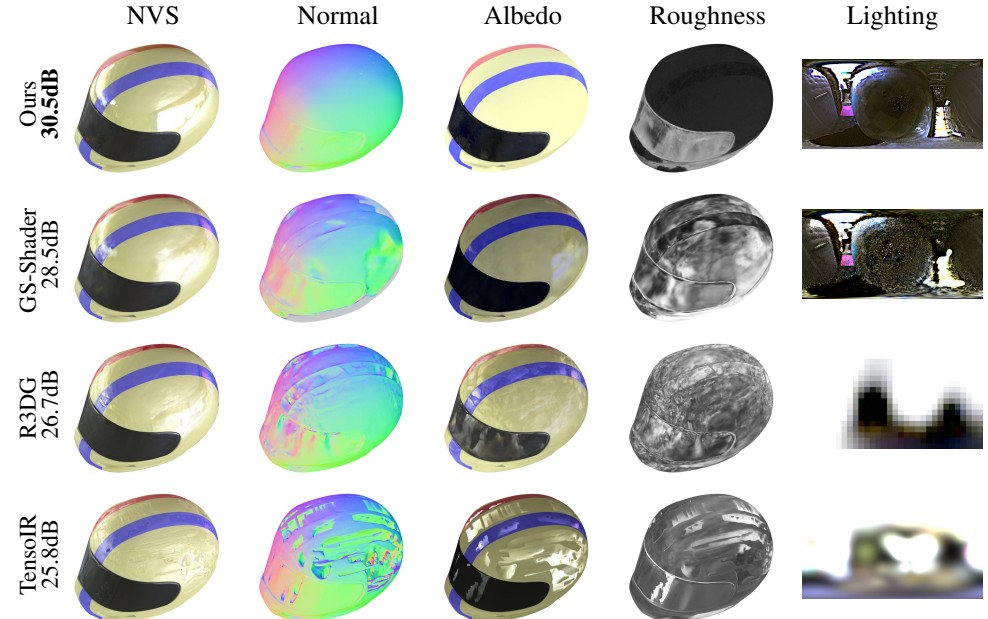

Figure 6: **Qualitative comparison on the Shiny Blender dataset**. Our method achieves the best decomposition performance for reflective objects.

## 4 EXPERIMENTS

We perform extensive experiments to verify the effectiveness of our inverse rendering method. We first evaluate decomposition ability in Sec. 4.1, demonstrating our superior inverse rendering performance on reflective cases. Next, we present material decomposition and relighting results of Synthetic4Relight in Sec. 4.2, showcasing our effectiveness in general inverse rendering tasks. Additionally, we report geometric reconstruction accuracy in Sec. 4.3, and ablation studies in Sec. 4.5. Further experiments are provided in Appendix. Our method is highly efficient, completing training within 20 minutes for the first stage and 3-5 minutes for the second stage—on a NVIDIA GTX 4090.

### 4.1 PERFORMANCE ON REFLECTIVE CASES

**Datasets & Metrics**. We evaluate NVS performance, normal quality and training efficiency on the Shiny Blender dataset, which includes 6 challenging scenes featuring complex lighting-material interactions. Qualitative results are provided in Fig. 6, highlighting the excellent efficiency and superior rendering quality among inverse rendering baselines, as shown in Table 1.

**Performance & Discussion**. We compare our method with a wide range of inverse rendering approaches. Our results achieve new state-of-the-art performance on reflective cases over all inverse rendering techniques as shown in Table 1. While achieving the best normal quality, GeoSplatting also outperform the second best GaussianShader almost 4dB in PSNR, demonstrating the effectiveness of our geometry guidance in accurately capturing high-frequency lighting-material interactions. Moreover, our method achieves significant superior optimization efficiency, where each scene takes only 20 minutes in average. Fig. 6 shows the qualitative NVS results, further demonstrating our effectiveness. More results on the Shiny Blender dataset are provided to Appendix H.3.

| Method | NVS (PSNR ↑) | Normal (MAE ↓) | Training Time (minutes ↓) |
|---|---|---|---|
| R3DG | 28.83 | 7.04 | ∼140 |
| TensoIR | 27.98 | 4.42 | ∼270 |
| NVdiffrec | 28.14 | 9.38 | 72 |
| GS-Shader | 30.64 | 7.03 | 63 |
| Ours | **34.41** | **2.70** | **20** |

Table 1: **Quantitative results on the Shiny Blender dataset.** In the left table, GeoSplatting significantly outperforms all baselines in NVS quality, normal accuracy, and training efficiency. In the right figure, we provide an intuitive overview, with the top-left corner representing superior efficiency. Notably, our method achieves excellent efficiency in terms of both training time and FPS.

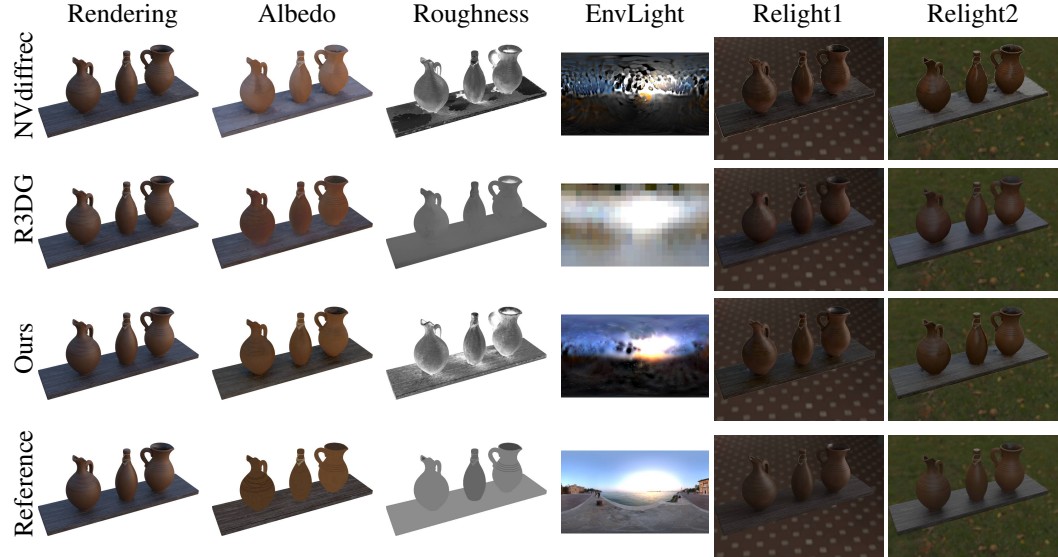

Figure 7: **Qualitative comparison of material decomposition & relighting on Synthetic4Relight Dataset**. Our method successfully recovers accurate albedo and lighting, though the roughness is slightly affected by indirect inter-reflections. Nevertheless, it still achieves the best relighting effects.

## 4.2 DECOMPOSITION & RELIGHTING PERFORMANCE ON GENERAL CASES

**Datasets & Metrics**. We evaluate Material decomposition and relighting performance on the Synthetic4Relight dataset (Zhang et al., 2022), which includes 4 challenging scenes with complex non-Lambertian materials and indirect lighting effects. Following (Gao et al., 2023), we assess NVS, Relighting, Albedo quality using PSNR, SSIM, and LPIPS metrics, along with Roughness MSE, as shown in Table 2. Qualitative results are provided in Fig. 7.

**Performance & Discussion.** We compare our method with mesh-based (NVdiffrec) and Gaussian-based relightable approaches (R3DG, GS-Shader, GS-IR). As shown in Table 2 and Fig. 7, while our method significantly outperforms the baseline methods in novel view synthesis, we also achieve state-of-the-art performance in relighting and albedo assessment, demonstrating superior material and lighting disentanglement. In terms of roughness estimation, our method performs on par with R3DG. In scenarios involving self-occlusion, our approach tends to integrate shadow effects into albedo or roughness. This behavior arises primarily from the limitations of the split-sum approximation used in our method. A detailed explanation is provided in Appendix D. To further clarify these observations, we include additional qualitative examples in Appendix H.2. Experimental results from the Shiny Blender dataset (Verbin et al., 2022b) and the TensoIR Synthetic dataset (Jin et al., 2023) are also presented in Appendices H.3 and H.4, respectively, to reinforce our findings and offer broader context.

| Method | Novel View Synthesis | | | Relighting | | | Albedo | | | Roughness |
|---|---|---|---|---|---|---|---|---|---|---|
| | PSNR ↑ | SSIM ↑ | LPIPS ↓ | PSNR ↑ | SSIM ↑ | LPIPS ↓ | PSNR ↑ | SSIM ↑ | LPIPS ↓ | MSE ↓ |
| NVdiffrec | 34.99 | 0.979 | 0.034 | 28.89 | 0.953 | 0.061 | 28.66 | 0.941 | 0.066 | 0.026 |
| GS-IR | 33.85 | 0.964 | 0.050 | 23.81 | 0.902 | 0.086 | 26.66 | 0.936 | 0.085 | 0.825 |
| GS-Shader | 30.26 | 0.974 | 0.029 | 22.32 | 0.924 | 0.084 | N/A | N/A | N/A | 0.050 |
| R3DG* | 36.80 | 0.982 | 0.028 | 31.00 | 0.964 | 0.050 | 28.31 | 0.951 | **0.058** | **0.013** |
| Ours | **39.20** | **0.988** | **0.013** | **31.65** | **0.971** | **0.032** | **29.21** | **0.952** | 0.062 | 0.017 |

Table 2: **Quantitative Results on the Synthetic4Relight Dataset** (* means copied from original papers). Our method achieves the best performance in NVS, relighting, and albedo, and on-par performance in roughness compared to R3DG. Note that GS-Shader does not provide disentangled albedo but rather a diffuse color merged with lighting, so we leave it as N/A. Also, note that we apply the albedo scaling introduced in (Zhang et al., 2021b) to perform a fair comparison.

## 4.3 PERFORMANCE ON GEOMETRY RECOVERY

**Datasets & Metrics**. We evaluate geometry recovery performance on four deliberately selected scenes: Spot, Damicornis, Lego, and Chair, each chosen for its distinct characteristics. Spot is highly reflective, Damicornis and Lego feature complex geometries, and Chair contains detailed

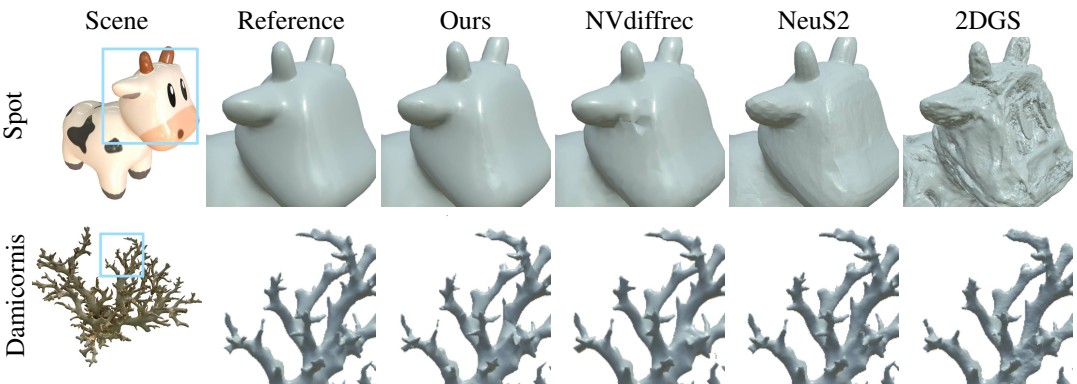

Figure 8: **Qualitative Geometry Comparison**. Our method achieves accurate geometry in scenes with challenging lighting and material conditions (shiny Spot) and complex topology (Damicornis).

textures. Following (Munkberg et al., 2022), we assess performance using the Chamfer distance and F-score, as shown in Table 3. Since our method includes meshes in stage 1 and Gaussian points in both stages 1 and 2, we provide metrics for each. Qualitative results are shown in Fig. 8.

**Performance & Discussion**. We compare our method with SDF-based (NeuS2), mesh-based (NVdiffrec), and Gaussian-based (2DGS) approaches. As demonstrated in Table 3 and Figure 8, our method shows robust performance across diverse scenarios. In highly reflective cases (Spot), NeuS2 and 2DGS struggle to accurately capture geometry due to their limited inverse rendering capabilities. For other scenes with more intricate geometry, our approach still achieves comparable geometry recovery performance to state-of-the-art surface reconstruction techniques, underscoring the effectiveness of our intermediate mesh guidance strategy. In Appendix H.3, we present a normal quality comparison on the Shiny Blender dataset (Verbin et al., 2022b) with challenging reflective scenarios, further highlighting how our geometry guidance enhances performance in inverse rendering tasks.

| Method | Spot | | Damicornis | | Lego | | Chair | |
|---|---|---|---|---|---|---|---|---|
| | Chamfer ↓ | F-score ↑ | Chamfer ↓ | F-score ↑ | Chamfer ↓ | F-score ↑ | Chamfer ↓ | F-score ↑ |
| NeuS2 | 12.43 | 0.9773 | **0.12** | 0.9986 | **7.03** | **0.9187** | 7.94 | 0.9046 |
| 2DGS | 39.13 | 0.6588 | 0.40 | **0.9993** | 13.23 | 0.9169 | 3.56 | **0.9594** |
| NVdiffrec | 1.04 | 0.9921 | 0.27 | 0.9974 | 11.38 | 0.8506 | 5.76 | 0.9359 |
| Ours (Stage1 Points) | 0.53 | 0.9995 | 16.56 | 0.9969 | 11.34 | 0.9027 | 4.20 | 0.9442 |
| Ours (Stage2 Points) | 0.49 | 0.9995 | 12.40 | 0.9980 | 8.71 | 0.9130 | **3.11** | 0.9570 |
| Ours (Mesh) | **0.16** | **0.9997** | 0.55 | 0.9985 | 9.03 | 0.8886 | 5.37 | 0.9254 |

Table 3: **Quantitative results on Geometry Recovery**. With each scene normalized to the range $[-1, 1]^3$, we report the Chamfer distance (scaled by $10^{-4}$) and the F-score (under a threshold of $10^{-3}$). While performing on par with other state-of-the-art methods in Lego, Chair and Damicornis, GeoSplatting achieves the best geometry in challenging reflective cases (Spot), thanks to its strong inverse rendering capabilities.

### 4.4 PERFORMANCE ON REAL-WORLD DATASET

**Datasets, Performance & Discussion**. Lastly, we show qualitative results on real-world DTU dataset (Aanæs et al., 2016). While GeoSplatting successfully decomposes reasonable geometry and material, as shown in Fig. 9, the real-world data is still much more challenging than synthetic data due to the inaccurate camera, complex lighting, and self occlusions. For instance, an overestimated roughness can be observed in Scan 65, mainly due to overexposure of input views. More failure cases on DTU dataset are provided in Appendix G.

### 4.5 ABLATION STUDIES

**Second Stage Optimization**. The second stage optimization plays a crucial role in improving the performance of novel view synthesis. On the NeRF Synthetic Dataset, it helps improve the PSNR from 29.52 to 32.32 (see Table 8 in Appendix H.1). The key issue lies in the isosurface's reliance on grid sampling ($96^3$ for FlexiCubes), which struggles to represent detailed geometry and textures. Therefore, in the second stage, we optimize the Gaussian representation without the constraints of the mesh, allowing it to fully express its representational power. Moreover, the good initialization in

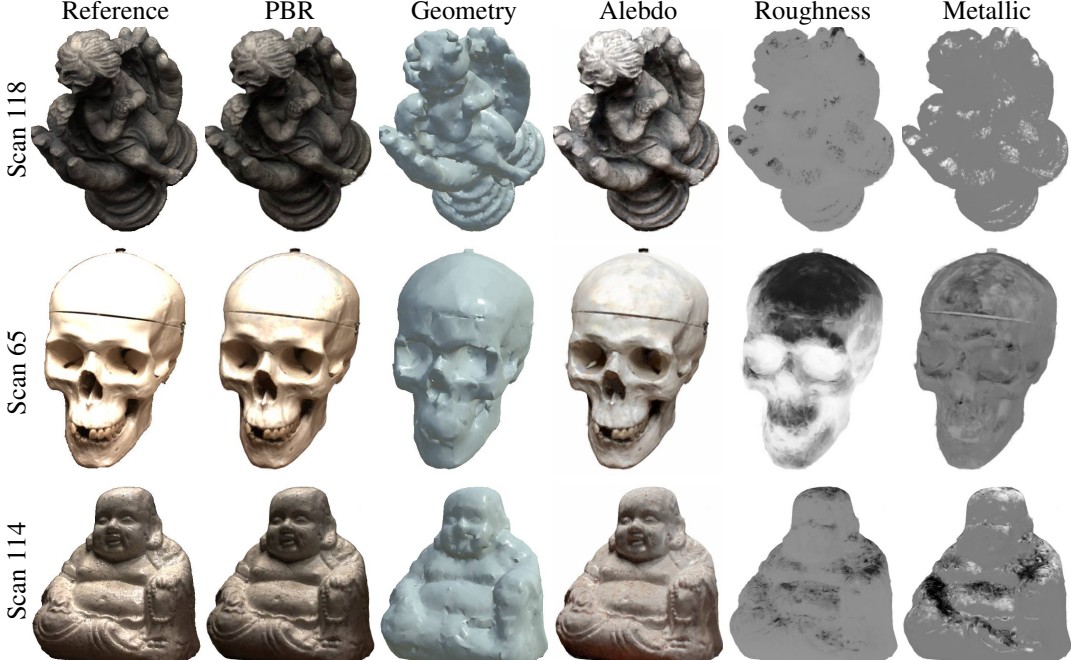

Figure 9: **Qualitative Results on DTU dataset**. Our method successfully recovers meaningful geometry and material on the challenging real-world dataset.

the first stage still guides the optimization toward meaningful decomposition. As shown in Fig. 10, the second stage significantly enhances the performance on normal leading to improved NVS results.

**Residual Terms**. We find it also helps improve the performance of novel view synthesis, *e.g.*, in the Chair scene, the PSNR drops by 2dB without residual terms. GeoSplatting applies a split-sum model to represent PBR lighting effects. While it achieves delicate decomposition results, it assumes a single-bounce rendering process, *i.e.*, light hits an object and reflects back to the light source, without considering any inter-reflection effects. The inclusion of residual terms significantly improves inverse rendering performance by attributing noise and higher-order lighting effects to the residual terms. As shown in Fig. 5, it successfully models the small inter-reflected green ball.

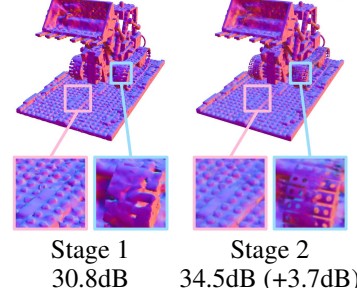

Stage 1          Stage 2
30.8dB        34.5dB (+3.7dB)

Figure 10: Second Stage Optimization on Lego.

## 5 CONCLUSION

**Limitation**: While GeoSplatting demonstrates state-of-the-art performance in NVS and relighting tasks, it still faces several challenges that motivate further research. First, its geometry guidance is derived from the iso-surface. Although this significantly improves the geometry performance of 3DGS, it also requires masks during training and is constrained by grid resolution, which limits its application to object-level inverse rendering tasks. A promising direction for future work would be to explore how to eliminate the need for masks and to apply adaptive resolution to accommodate detailed geometry, enabling its extension to scene-level tasks. Furthermore, GeoSplatting currently models only single-bounce specular lighting, leaving the higher-order effects (*e.g.*, inter-reflections) to residual terms. Shadows will be baked into albedo as well, resulting in an inaccurate appearance under relighting conditions. However, since we have access to intermediate meshes, incorporating ray tracing techniques could enable a more comprehensive decomposition of shadows and inter-reflections. These areas hold great potential, and we aim to explore them in future work.

**Conclusion**: We propose GeoSplatting, a novel hybrid representation that enhances 3DGS with explicit geometric guidance and differentiable PBR equations. GeoSplatting achieves superior efficiency and state-of-the-art inverse rendering performance in reflective scenes, which demonstrates its impressive ability in modeling high-frequency lighting-material interaction. Additionally, for general cases, GeoSplatting delivers decomposition results that are comparable to those of state-of-the-art inverse rendering baselines. We will release all the code to facilitate related research.

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

## A  APPENDIX

In the appendix, we provide a comprehensive explanation about the details of our work, including detailed implementations and limitations of our method, as well as supplementary results from both quantitative and qualitative experiments.

The appendix begins with a detailed explanation of the MGadapter in Appendix B, followed by an overview of the implementation details for our loss functions in Appendix C. In Appendix D, we analyze two approaches for PBR computation (split-sum and spherical Gaussian) before examining the impact of different FlexiCubes resolutions in Appendix E. To explore the limitations of our method, we include an ablation study on the input mask required during training in Appendix F, as well as an analysis of failure cases in Appendix G. Finally, we present additional results from novel view synthesis (NVS) and relighting experiments in Appendix H.

## B  EXPLANATION OF MGADAPTER

### B.1  OVERVIEW

We first describe the details of our MGadapter. As discussed in Sec. 3.1, the MGadapter takes a triangle mesh as input and generates a set of Gaussian points corresponding to the mesh's shape. The core idea of MGadapter is to ensure that the shape of the triangle mesh aligns with that of the Gaussian points, serving as a differentiable adapter between the two. However, since Gaussian points lack the discrete geometric boundaries present in meshes, we define geometric alignment as follows: given a triangle face, we can sample several viewpoints and render the triangle from them. The geometric alignment is then measured by the difference between the rendered mesh and the rendered Gaussian points (e.g. $L_1$ loss on depth maps).

The intuitive implementation of our MGadapter involves sampling several Gaussian points on the mesh surface. These points are then optimized in terms of scale and rotation to minimize the depth map difference between the Gaussian points and the target mesh. However, this approach introduces an additional optimization step that must be re-executed each time the mesh is modified, resulting in reduced optimization efficiency and an unstable training process.

Instead of performing geometric alignment in real-time, we propose utilizing a predefined heuristic function $\mathcal{T}$ to achieve an approximate alignment. As described in Eq. 1, the MGadapter $\mathcal{T}$ takes arbitrary triangle meshes as input and outputs Gaussian point attributes, including position $\boldsymbol{\mu}$, scale $\mathbf{S}$, rotation $\mathbf{R}$, and normal $\mathbf{n}$. This process acts as a generalized adapter between the input meshes and the corresponding Gaussian points. In Appendix B.2, we provide a detailed explanation of Eq. 1. Then, we discuss the implementation of the surface adjustment in Appendix B.4, which is critical for Gaussian point rendering. Finally, we explain how to query Gaussian point attributes from a spatial MLP in Appendix B.5 and explain the warm-up stage in Appendix B.3.

### B.2  EXPLANATION OF EQ. 1

Specifically, given the triangle mesh, each triangle face $F_i$ comprises three vertices $P_i = (p_{i1}, p_{i2}, p_{i3})$ with their vertex normals $N_i = (n_{i1}, n_{i2}, n_{i3})$. We symmetrically sample 6 points on $F_i$ with barycentric coordinates:

$$
\begin{aligned}
&b_1 = (u, u, 1-2u) \quad b_2 = (u, 1-2u, u) \quad b_3 = (1-2u, u, u) \\
&b_4 = (v, v, 1-2v) \quad b_5 = (v, 1-2v, v) \quad b_6 = (1-2v, v, v)
\end{aligned}
\tag{7}
$$

And we can obtain 6 midpoints $m_{jk}$:

$$
\begin{aligned}
m_{12} &= \frac{b_1 + b_2}{2} = \left(u, \frac{1-u}{2}, \frac{1-u}{2}\right) \quad m_{45} = \frac{b_4 + b_5}{2} = \left(v, \frac{1-v}{2}, \frac{1-v}{2}\right) \\
m_{23} &= \frac{b_2 + b_3}{2} = \left(\frac{1-u}{2}, \frac{1-u}{2}, u\right) \quad m_{56} = \frac{b_5 + b_6}{2} = \left(\frac{1-v}{2}, \frac{1-v}{2}, v\right) \\
m_{31} &= \frac{b_3 + b_1}{2} = \left(\frac{1-u}{2}, u, \frac{1-u}{2}\right) \quad m_{64} = \frac{b_6 + b_4}{2} = \left(\frac{1-v}{2}, v, \frac{1-v}{2}\right)
\end{aligned}
\tag{8}
$$

Given an attribute $A_i = (a_{i1}, a_{i2}, a_{i3})$ defined at the triangle vertices, we represent the barycentric interpolation as:

$$(b_1, b_2, b_3) \odot A_i = b_1 a_{i1} + b_2 a_{i2} + b_3 a_{i3} \tag{9}$$

Then, for each midpoint $m_{jk}$ ($jk = 12, 23, 31, 45, 56, 64$), we sample a Gaussian point as:

$$
\begin{aligned}
\boldsymbol{\mu} &= m_{jk} \odot P_i & \mathbf{n} &= m_{jk} \odot N_i \\
\mathbf{S}_x &= \alpha_{jk} \| b_k \odot P_i - m_{jk} \odot P_i \|_2 & \mathbf{R}_x &= \frac{b_k \odot P_i - m_{jk} \odot P_i}{\| b_k \odot P_i - m_{jk} \odot P_i \|_2} \\
\mathbf{S}_y &= \frac{\text{Area}(F_i)}{\beta_{jk} \| b_k \odot P_i - m_{jk} \odot P_i \|_2} & \mathbf{R}_y &= \mathbf{n} \times \mathbf{R}_x \\
\mathbf{S}_z &= \delta_{jk} & \mathbf{R}_z &= \mathbf{n}
\end{aligned}
\tag{10}
$$

Here, Eq. 10 provide the formulation of our heuristic function $\mathcal{T}$, with $u, v, \alpha_{jk}, \beta_{jk}, \delta_{jk}$ as hyper-parameters. To achieve the generalized geometric alignment, we practically set these parameters as follows:

$$
\begin{aligned}
u &= 0.07 \\
v &= 0.22 \\
\alpha_{12} = \alpha_{23} = \alpha_{31} &= 0.80 \\
\alpha_{45} = \alpha_{56} = \alpha_{64} &= 2.08 \\
\beta_{12} = \beta_{23} = \beta_{31} &= 15.0 \\
\beta_{45} = \beta_{56} = \beta_{64} &= 13.0 \\
\delta_{12} = \delta_{23} = \delta_{31} = \delta_{45} = \delta_{56} = \delta_{64} &= 4.5 \times 10^{-5}
\end{aligned}
\tag{11}
$$

### B.3 EXPLANATION OF WARM-UP STAGE

Next, we explain the warm-up stage during training. As outlined in Appendix B.2, we typically sample six Gaussian points from each triangle surface. However, during the initial training stage when the underlying mesh has not yet converged, there can be an excessive number of triangle slices, as illustrated in Fig. 11. Sampling six Gaussian points per face in this scenario can result in significant memory costs and reduced training efficiency.

To address this issue, we implement a warm-up stage for MGadapter at the beginning of training (covering the first 2%). During this phase, MGadapter outputs a significantly reduced number of Gaussian points by performing vertex sampling. Specifically, MGadapter samples a single Gaussian point from each mesh vertex and assigns its normal to match the corresponding vertex normal. For the scales and rotations of the sampled Gaussian points, we utilize two small spatial MLPs to learn these attributes. Once the warm-up stage concludes, the two spatial MLPs are discarded, and the scales and rotations are afterwards computed as described in Appendix B.2.

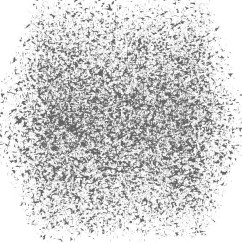

Figure 11: Initial mesh slices of FlexiCubes.

### B.4 EXPLANATION OF SURFACE ADJUSTMENT

By applying Eq. 10, we obtain a set of Gaussian points that lie exactly on the surface. However, as mentioned in Sec. 3.1, strictly positioning the Gaussian points on the surface can negatively impact rendering quality, particularly near the boundaries between distinct texture colors.

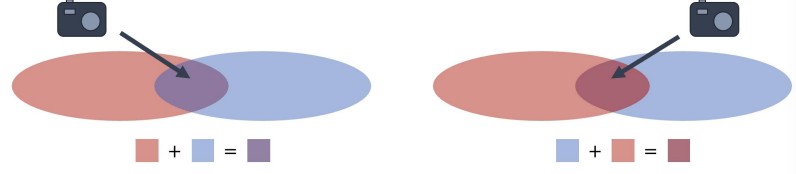

Figure 12: **Inconsistent Sorting.** The vanilla 3D splatting algorithm produces multi-view-inconsistent depths, leading to varying blending sequences across different views.

This issue primarily arises from the approximations made during the projection transformation of 3DGS, as noted in (Zwicker et al., 2004). These approximations lead to multi-view-inconsistent sorting in the overlapping regions between two surface-aligned Gaussian points, as illustrated in Fig. 12. Consequently, Gaussian points near color boundaries struggle to learn a consistent appearance. Fig. 13(a) presents rendering results when Gaussian points are strictly positioned on the surface, further demonstrating this problem. However, upon noticing that vanilla 3DGS can achieve high rendering quality despite the projection approximations, we perform further analysis and found it automatically

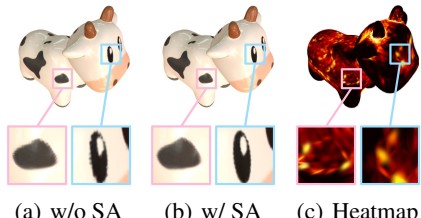

(a) w/o SA    (b) w/ SA    (c) Heatmap

Figure 13: **Surface Adjustment (SA) Explanation.** (a) PSNR: 31.9dB; (b) PSNR: 33.5dB; (c) Magnitude of SA.

learns to position those boundary Gaussians deeper, placing them beneath the actual surface. This adjustment results in a consistent depth sorting. Therefore, in our MGadapter, we also allow Gaussian points to learn a small offset along the normal direction. As demonstrated in Fig. 13(b), such surface adjustment can significantly enhance rendering quality, which is important for modeling reflective surface accurately in inverse rendering tasks. The magnitude of this surface adjustment (measured by $\|v\|$) is visualized in Fig. 13(c), highlighting larger adjustments near the color boundaries.

The depth error from the projection approximation in 3DGS has also been discussed in recent work, specifically 2DGS (Huang et al., 2024). Rather than applying 3D splatting to flat Gaussian points, 2DGS employs a ray-splat intersection algorithm to ensure depth-precise rendering, resulting in a view-consistent appearance. However, when integrating the 2D splatting algorithm into our pipeline without surface adjustment, we observe strong floater artifacts, which are illustrated in Fig. 14, probably due to the incompatibility between 2DGS and FlexiCubes. Specifically, as noted in the original paper, 2D Gaussian points can degenerate into a line when observed from a slanted viewpoint. In this context, 2DGS renders these Gaussian points differently, and we empirically find it causes FlexiCubes to struggle with reducing mesh slices shown in Fig. 11, leading to unwanted floaters. We conducted a quantitative analysis comparing the two splatting algorithms, with results presented in Table 4, demonstrating the incompatibility between 2DGS and FlexiCubes. Consequently, we have chosen to employ 3D splatting instead.

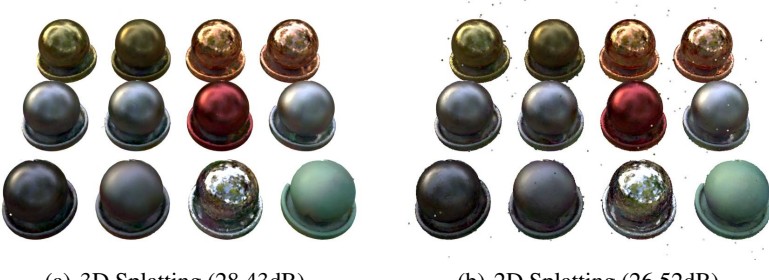

(a) 3D Splatting (28.43dB)        (b) 2D Splatting (26.52dB)

Figure 14: **Qualitative Comparison between 2D Splatting and 3D Splatting.** Producing incorrect rendering results for tiny surfels, 2DGS encounters challenges in reducing the triangle slices initially generated by FlexiCubes, leading to noticeable floaters.

| Method | Chair | Drums | Ficus | Hotdog | Lego | Materials | Mic | Ship | Avg. |
|---|---|---|---|---|---|---|---|---|---|
| Ours (3D splatting) | 31.98 | 24.53 | 28.96 | 33.85 | 30.83 | 28.43 | 31.32 | 26.23 | 29.52 |
| Ours (2D splatting) | 29.44 | 23.71 | 26.34 | 31.18 | 29.86 | 26.52 | 28.21 | 25.22 | 27.56 |

Table 4: Splatting algorithm comparison on NeRF dataset (PSNR↑).

## B.5 EXPLANATION OF SPATIAL MLP

Lastly, we discuss how to model PBR attributes and surface adjustments on Gaussian points. Since Gaussian points are generated in real time from the underlying mesh, directly modeling these at-

tributes as learnable parameters is impractical due to the varying number of Gaussian points during training. Instead, we employ a spatial MLP $F$ to construct an attribute field, as mentioned in Sec. 3.3.

Specifically, the spatial MLP incorporates the multi-resolution hash encoding introduced in (Müller et al., 2022), followed by a compact MLP. For any spatial coordinate $p \in [-1, 1]^3$, the spatial MLP outputs $F(p) \in \mathbb{R}^C$. Implemented using tiny-cuda-nn (Müller, 2021), we utilize two spatial MLPs, $F_{\text{PBR}}$ and $F_{\text{SA}}$, to model PBR attributes and surface adjustments, respectively. Detailed parameters can be found in Table 5.

| Module | Parameter | Value |
|---|---|---|
| | Number of levels | 16 |
| | Max.entries per level (hash table size) | $2^{19}$ |
| $F_{\text{PBR}}/F_{\text{SA}}$ HashEnc | Number of feature dimensions per entry | 2 |
| | Coarsest resolution | 32 |
| | Finest resolution | 4096 |
| | MLP layers | $32 \times 32 \times 32 \times 6$ |
| $F_{\text{PBR}}$ MLP | Initialization | Kaiming-uniform |
| | Final activation | Sigmoid |
| | MLP layers | $32 \times 32 \times 1$ |
| $F_{\text{SA}}$ MLP | Initialization | Kaiming-uniform |
| | Final activation | None |

Table 5: Parameters of Spatial MLP

## C    DETAILS OF LOSS FUNCTIONS

### C.1    PHOTOMETRIC TERM

During the training stage, for each view $i$, GeoSplatting differentiably renders a RGB image $I_{pred}^{(i)} \in \mathbb{R}^{H \times W \times 3}$ and takes the alpha channel as the mask $M_{pred}^{(i)} \in \mathbb{R}^{H \times W \times 1}$. Given the ground truth $I_{gt}^{(i)}$ and $M_{gt}^{(i)}$ for view $i$, the photometric loss is computed as:

$$
\begin{aligned}
\mathcal{L}_{photo} &= \mathcal{L}_1 + \lambda_{ssim}\mathcal{L}_{SSIM} + \lambda_{mask}\mathcal{L}_{mask} \\
&= \|I_{gt}^{(i)} - I_{pred}^{(i)}\|_1 + \lambda_{ssim}\text{SSIM}(I_{gt}^{(i)}, I_{pred}^{(i)}) + \lambda_{mask}\|M_{gt}^{(i)} - M_{pred}^{(i)}\|_2
\end{aligned}
\tag{12}
$$

Here, $\lambda_{ssim} = 0.2$ and $\lambda_{mask} = 5.0$ for all the cases.

### C.2    ENTROPY REGULARIZATION TERM

Following DMTet and FlexiCubes (Shen et al., 2021; 2023), we add an entropy loss to constrain the shape. Specifically, we employ FlexiCubes as the underlying geometric representation, which defines a scalar function $\zeta : \mathbb{R}^3 \to \mathbb{R}$ on the underlying cube grids $\mathcal{G}(\mathcal{V}, \mathcal{E})$ and then extracts isosurfaces via the differential Dual Marching Cubes introduced by (Shen et al., 2023). Given an edge $(v_i, v_j)$ from edge set $\mathcal{E}$, the SDF values defined on the endpoints $v_i, v_j$ are respectively $\zeta(v_i)$ and $\zeta(v_j)$.

Then, we can compute the regularization term as:

$$
\mathcal{L}_{sdf} = \sum_{(v_i, v_j) \in \mathcal{E}, \text{sgn}(\zeta(v_i)) \neq \text{sgn}(\zeta(v_j))} \mathcal{H}(\zeta(v_i), \text{sgn}(\zeta(v_j))) + \mathcal{H}(\zeta(v_j), \text{sgn}(\zeta(v_i)))
\tag{13}
$$

Here, $\mathcal{H}$ denotes the binary cross entropy. By encouraging the same sign of $\zeta$, such a regularization term penalize internal geometry and floaters.

### C.3    SMOOTHNESS REGULARIZATION TERM

Following NVdiffrec and R3DG (Munkberg et al., 2022; Gao et al., 2023), we apply smoothness regularization on albedo, roughness, and metallic to prevent dramatic high-frequency variations.

Given the positions $\boldsymbol{\mu}$ of Gaussian points, the albedo, roughness and metallic attributes are generated by the spatial MLP:

$$(v, k_d, r, m, c^r) = F(\boldsymbol{\mu}) \tag{14}$$

While applying a small perturbation on $\boldsymbol{\mu}$ can yield a different set of attributes:

$$\left(v', k_d', r', m', c^{r'}\right) = F(\boldsymbol{\mu} + \Delta\boldsymbol{\mu}) \tag{15}$$

The smoothness are computed as:

$$\mathcal{L}_{smooth} = \lambda_{albedo}\|k_d - k_d'\|_1 + \|r - r'\|_1 + \|m - m'\|_1 \tag{16}$$

Here, $\lambda_{albedo} = 6.0$.

### C.4    Light Regularization Term

Following NVdiffrec and R3DG (Munkberg et al., 2022; Gao et al., 2023), we add white balance regularization to prevent the albedo from being baked into the environment map.

Given a learnable environment map $L \in \mathbb{R}^{6 \times 512 \times 512 \times 3}$ which can be split into RGB channels $L_R, L_G, L_B \in \mathbb{R}^{6 \times 512 \times 512}$, the white balance regularization is computed as:

$$\mathcal{L}_{light} = \frac{1}{3}\left(\|L_R - L_W\|_1 + \|L_G - L_W\|_1 + \|L_B - L_W\|_1\right) \tag{17}$$

where $L_W = \frac{1}{3}(L_R + L_G + L_B)$.

### C.5    Final Loss

The final loss $\mathcal{L}$ is computed as:

$$\mathcal{L} = \mathcal{L}_{photo} + \lambda_{sdf}\mathcal{L}_{sdf} + \lambda_{smooth}\mathcal{L}_{smooth} + \lambda_{light}\mathcal{L}_{light} \tag{18}$$

Here, $\lambda_{sdf}$ is initially set to 0.2 at the start of the training stage and is linearly decreased to 0.01 by the midpoint of the training. As for $\lambda_{smooth}$ and $\lambda_{light}$, typical settings are $\lambda_{smooth} = 0.005, \lambda_{light} = 0.0005$. For highly specular objects, $\lambda_{light}$ should be set to a smaller value, such as $0.00001$.

## D    Split-Sum Approximation vs. Spherical Gaussian

The physically-based rendering (PBR) equation, incorporating the GGX microfacet model, is expressed in Eq. 2 and Eq. 3. However, solving this equation requires extensive Monte Carlo sampling, which is computationally intensive and impractical for both real-time forward rendering and inverse rendering. To address this challenge, several efficient methods for approximating the PBR equation have been proposed. The two most widely adopted approaches are the split-sum approximation (Karis, 2013) and Spherical Gaussian (SG) representations (Chen et al., 2021). While the Spherical Gaussian approximations are commonly used by inverse rendering methods such as InvRender, TensoIR, and R3DG, methods like NVdiffrec, GaussianShader, and our GeoSplatting favor the split-sum approximation due to its efficiency and ability in modeling complicated lighting.

$$L_o(\mathbf{x}, \boldsymbol{\omega}_o) = \int_{\mathcal{H}^2} f_r(\mathbf{x}, \boldsymbol{\omega}_i, \boldsymbol{\omega}_o)L_i(\mathbf{x}, \boldsymbol{\omega}_i)|\mathbf{n} \cdot \boldsymbol{\omega}_i|d\boldsymbol{\omega}_i \tag{19}$$

$$= \int_{\mathcal{H}^2} f_r(\mathbf{x}, \boldsymbol{\omega}_i, \boldsymbol{\omega}_o)(L_{ind}(\mathbf{x}, \boldsymbol{\omega}_i) + V(\mathbf{x}, \boldsymbol{\omega}_i)L_{env}(\boldsymbol{\omega}_i))|\mathbf{n} \cdot \boldsymbol{\omega}_i|d\boldsymbol{\omega}_i \tag{20}$$

To improve computational tractability, Eq. 19 can be reformulated as Eq. 20, where the incoming illumination $L_i(\mathbf{x}, \boldsymbol{\omega}_i)$ is split into two components: the indirect lighting term $L_{ind}(\mathbf{x}, \boldsymbol{\omega}_i)$ and the environment lighting term $L_{env}(\boldsymbol{\omega}_i)$. This separation allows the PBR equation to be computed in two parts, the environment lighting term $L_{env}(\boldsymbol{\omega}_i)$ and the light transfer terms $L_{ind}(\mathbf{x}, \boldsymbol{\omega}_i)$ & $V(\mathbf{x}, \boldsymbol{\omega}_i)$, each interacting with the bidirectional reflectance distribution function (BRDF).

The split-sum approximation and Spherical Gaussian methods address different challenges in rendering complex lighting and light transfer scenarios. While both approaches have strengths, they also exhibit notable limitations.

1. The split-sum approximation excels in efficiently modeling complex environmental lighting. Its computational simplicity makes it a practical choice for scenarios where speed is critical. However, it struggles to accurately handle visibility terms. In highly occluded scenes, the errors introduced by the approximation in Eq. 4 become significant.

2. In contrast, the Spherical Gaussian approach effectively captures intricate light transfer phenomena such as inter-reflection and ambient occlusion. However, its capacity to model direct lighting is limited to low-frequency components. This restriction results in inaccuracies when dealing with highly specular surfaces, where precise decomposition is essential.

Despite the challenges posed by shadow and self-occlusion effects, as shown in Fig. 7, we choose to utilize the split-sum approximation for several compelling reasons:

1. **Efficiency in Training and Inference:** The SG-based methods integrated with ray tracing, both neural-field-based TensoIR and 3DGS-based R3DG, results in longer training times and lower inference frame rates, as highlighted in Fig. 15.

2. **Handling Reflective Cases:** SG-based methods struggle to accurately model complex lighting-material interactions in reflective scenarios, leading to subpar decomposition results, as discussed in Appendix D.1.

3. **Addressing Occlusion Challenges:** While highly occluded cases present difficulties, the inclusion of a learnable occlusion term can mitigate these issues, as detailed in Appendix D.2.

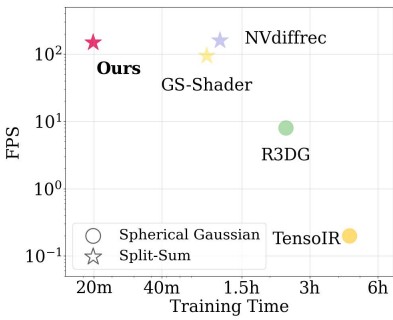

Figure 15: Comparison of Efficiency between Split-Sum and SG.

## D.1 REFLECTIVE CASES

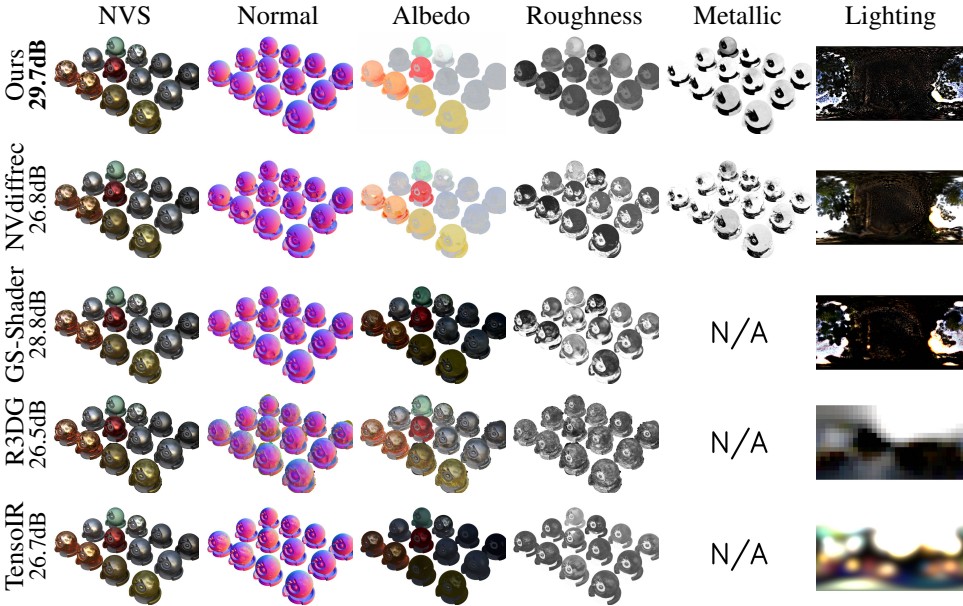

Figure 16: Incorrect Decomposition for SG-based Methods

The SG-based methods, such as TensoIR and R3DG, face significant challenges under complex lighting conditions, particularly when dealing with intricate material properties. As shown in Fig. 16, while split-sum-based methods achieve reasonable decomposition, TensoIR and R3DG fail to accurately reconstruct geometry, materials, and lighting. In Fig. 31, we provide qualitative decomposition results on the Shiny Blender dataset (Verbin et al., 2022b), to further demonstrate the strengths of our method on reflective cases.

## D.2 LEARNABLE OCCLUSION

Although all split-sum-based methods, including NVdiffrec and GS-Shader, struggle with self-occluded cases, it is still possible to mitigate the occlusion issue by modeling the visibility term $V(\mathbf{x}, \boldsymbol{\omega}_i)$ as described in Eq. 20. By introducing a learnable attribute $o_i$ for each Gaussian point $i$, we reformulate Eq. 5 as follows:

$$c_i = (1 - o_i)(c_i^d + c_i^s) + c_i^r \tag{21}$$

Here, $c_i^d$, $c_i^s$ and $c_i^r$ represent diffuse, specular, and residual terms, respectively. This learnable occlusion term improves our method's ability to handle shadow effects. As shown in 17, for the Hotdog scene from the Synthetic4Relight dataset and the TensoIR Synthetic dataset, with weaker shadow effects, our occlusion term effectively aids in albedo-shadow decoupling.

Note that despite the roughness quality on the Hotdog scene remaining influenced by shadow effects, the albedo quality of our method achieves the best among the baselines, as evidenced by Table 12 in Sec. H.4. However, for the Hotdog scene from the NeRF Synthetic dataset, which features strong shadow effects, our method, along with other baselines, struggles with strong shadow effects, as illustrated in Fig. 18. This common failure underscores shadow modeling as a persistent challenge in inverse rendering.

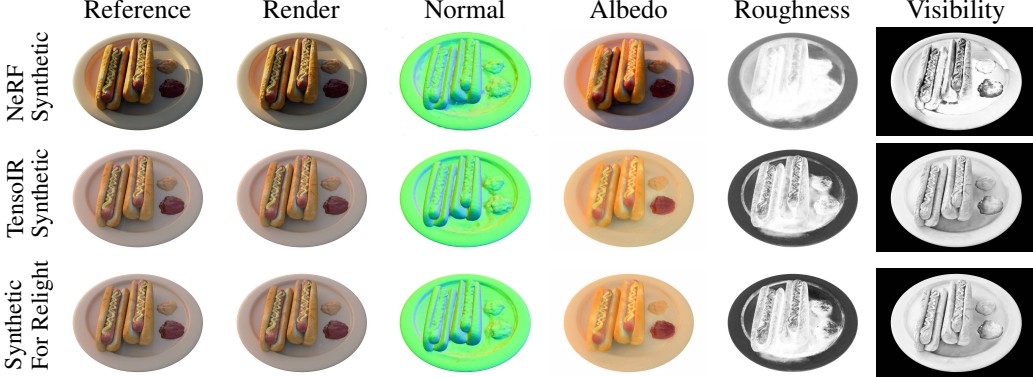

Figure 17: Qualitative comparison of decomposition performance under varying shadow effects.

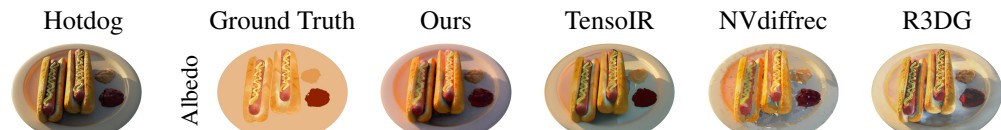

Figure 18: Qualitative results of albedo reconstruction performance under strong shadow effects.

## E ANALYSIS OF FLEXICUBES RESOLUTION

In this section, we examine the impact of FlexiCubes' grid resolution on our decomposition performance, focusing on rendering quality, albedo accuracy, normal quality, and training efficiency. Quantitative results are provided in Table 6, while qualitative results are shown in Fig. 19. Although FlexiCubes with higher resolution offer more precise normal guidance, the increased computational cost and inefficient training time may not justify the benefits.

| GeoSplatting | Novel View Synthesis | | | Albedo | | | Normal | Training Time |
|---|---|---|---|---|---|---|---|---|
| Resolution | PSNR ↑ | SSIM ↑ | LPIPS ↓ | PSNR ↑ | SSIM ↑ | LPIPS ↓ | MAE ↓ | minute(s)↓ |
| 64 | 36.31 | 0.9785 | 0.014 | 21.49 | 0.8799 | 0.128 | 8.451 | 14 |
| 96 | 37.15 | 0.9811 | 0.011 | 21.78 | 0.8824 | 0.118 | 7.880 | 24 |
| 128 | 37.57 | 0.9827 | 0.009 | 21.92 | 0.8825 | 0.117 | 7.313 | 96 |

Table 6: Impact of Grid Resolution on Decomposition Performance.

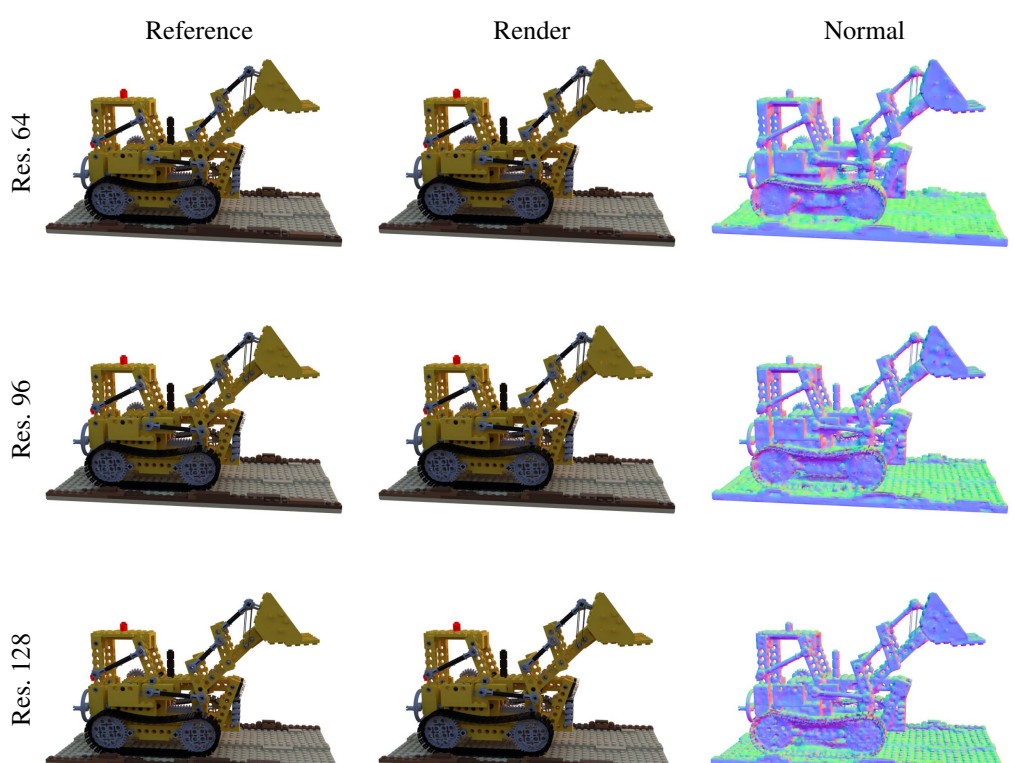

Figure 19: Qualitative Comparison of Different Grid Resolution.

## F    ANALYSIS OF MASK

As discussed in Appendix C.1, our method requires input masks during the training stage. Specifically, the mask term of the loss function in Eq. 12 is defined by the difference between the input masks and our predicted masks. This dependency introduces a limitation, as real-world data must first be segmented into foreground and background. To provide a comprehensive understanding, we present an ablation study to illustrate how the dependency on input masks varies across different scenes.

### F.1    OBJECT-LEVEL

Table 7 presents the performance differences in novel view synthesis (measured in terms of PSNR) on the NeRF Synthetic Dataset, while qualitative comparison are shown in Fig. 20. The results indicate that without the mask, our method has difficulty reconstructing smooth, convex surfaces with specular highlights, as seen in Materials and Mic. Additionally, for objects with thin structures, such as Ficus, performance significantly declines in the absence of the mask loss term. In contrast, for the other five objects in the NeRF Synthetic Dataset, the ground truth mask is not essential for achieving satisfactory results.

| Method | Chair | Drums | Ficus | Hotdog | Lego | Materials | Mic | Ship | Avg. |
|---|---|---|---|---|---|---|---|---|---|
| Ours w/ mask (stage 1) | 31.98 | 24.53 | 28.96 | 33.85 | 30.83 | 28.43 | 31.32 | 26.23 | 29.52 |
| Ours w/o mask (stage 1) | 31.92 | 23.53 | 26.11 | 33.95 | 31.02 | 24.81 | 30.99 | 26.08 | 28.55 |
| Difference | -0.06 | -1.00 | -2.85 | +0.10 | +0.19 | -3.62 | -0.33 | -0.15 | -0.97 |
| Ours w/ mask (stage 2) | 34.71 | 26.05 | 33.48 | 36.40 | 34.47 | 29.66 | 34.62 | 29.17 | 32.32 |
| Ours w/o mask (stage 2) | 34.81 | 25.72 | 31.56 | 36.36 | 34.73 | 28.22 | 34.31 | 28.54 | 31.78 |
| Difference | +0.10 | -0.33 | -1.92 | -0.04 | +0.26 | -1.44 | -0.31 | -0.63 | -0.54 |

Table 7: Quantitative results of mask ablation study (PSNR↑).

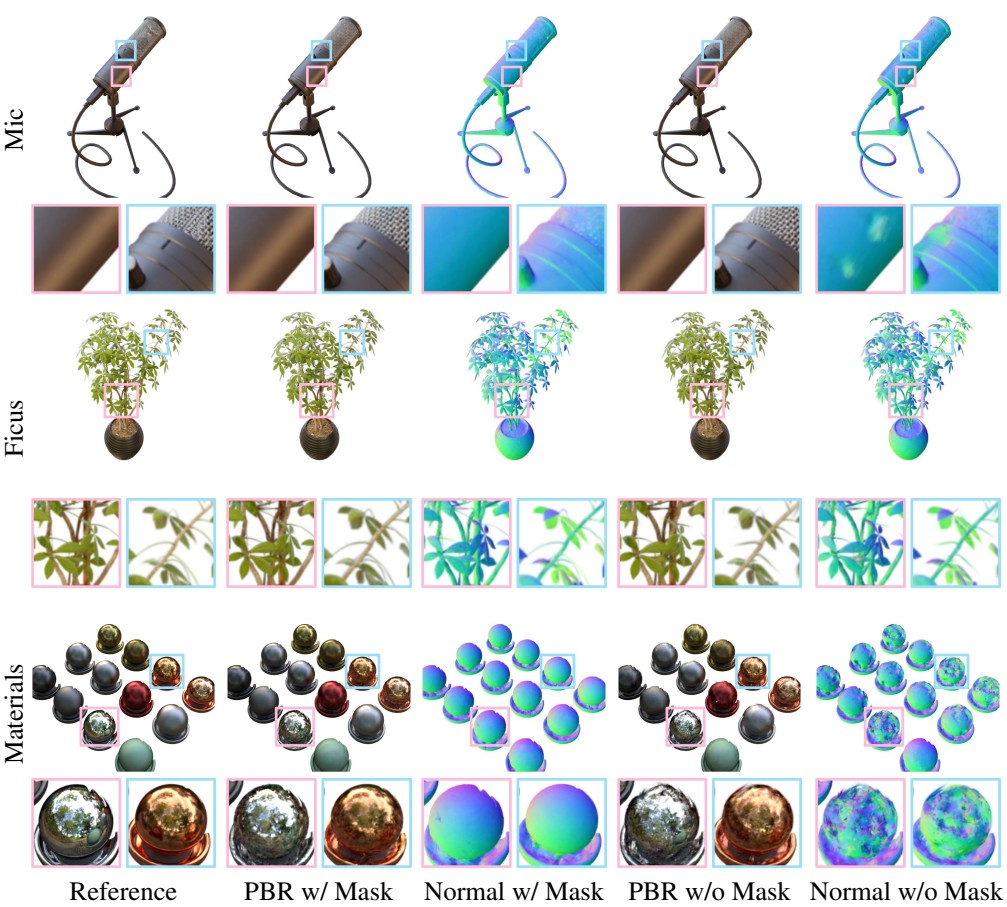

Figure 20: Qualitative results of mask ablation study.

### F.2 SCENE-LEVEL

Without input masks, our method will completely fail on scene-level cases, as illustrated in Fig. 21. To extend our method from object-level decomposition to scene-level decomposition, a promising direction is to explore how to eliminate the need for masks and to apply adaptive resolution to accommodate detailed geometry.

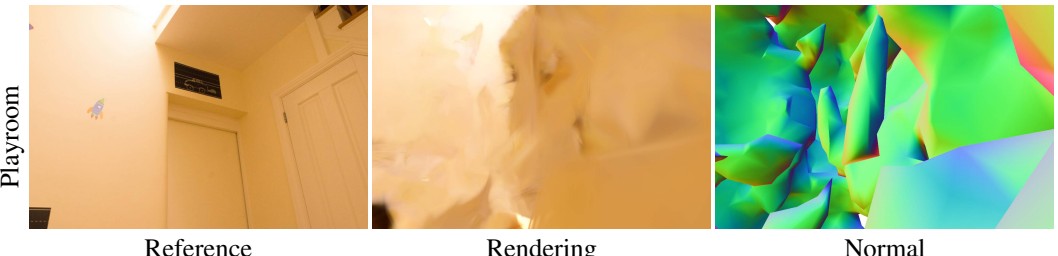

Figure 21: Failure on scene-level decomposition tasks.

## G FAILURE CASES

We present a series of failure cases to illustrate the limitations of our method. The qualitative examples from both the synthetic dataset and the DTU Dataset highlight scenarios that lead to incorrect decomposition or poor geometry.

### G.1 THIN STRUCTURES

As discussed in Sec. 4.5, our method struggles with thin structures in the first stage due to grid resolution limitations. While the second stage relaxes positional constraints on Gaussian points to aid in recovering fine geometry, it still cannot perfectly reconstruct thin structures due to the absence of geometric guidance in Stage 2. Fig. 22 showcases failure cases involving the Ficus, Ship, and Air Balloons.

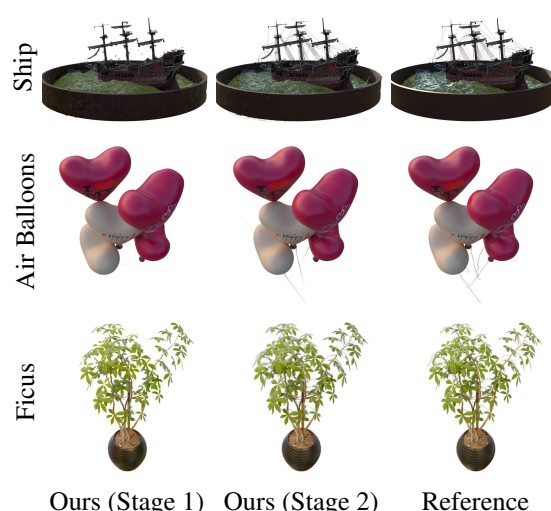

Ours (Stage 1)    Ours (Stage 2)    Reference

Figure 22: Failure cases of thin structures.

### G.2 INCONSISTENT LIGHTING

Variations in illumination conditions (e.g. exposure and shadows) across multiple views can lead to inconsistent lighting, especially for datasets captured in real-world environments. An illustrative example from DTU Dataset is provided in Fig. 23, which demonstrates significant illumination changes between View 38 and View 40. Consequently, our method can produce incorrect decompositions in these scenes, resulting in overestimated metallic, noisy lighting, and distorted geometry near the inconsistent regions, shown in Fig. 24.

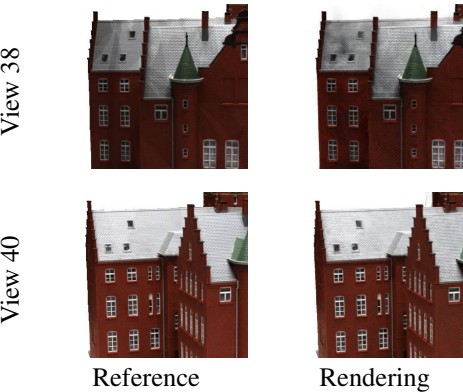

Reference          Rendering

Figure 23: Inconsistent lighting on Scan 24.

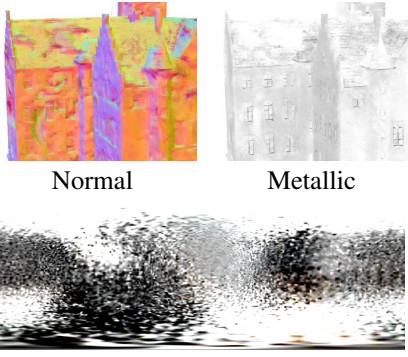

Normal            Metallic

EnvLight

Figure 24: Incorrect decomposition.

### G.3 UNDEREXPOSURE

Fig. 25 also illustrates a failure case in which the reference image is heavily underexposed. The ambiguous material-lighting composition in this scenario results in incorrect geometry recovery, as our method optimizes both aspects jointly.

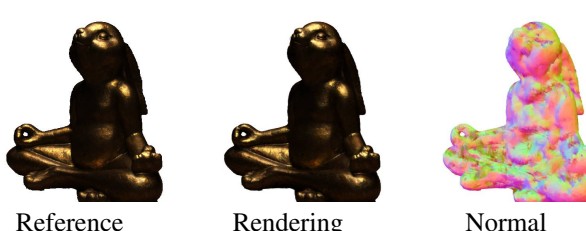

Reference          Rendering          Normal

Figure 25: Underexposed views from DTU Scan 110.

# H MORE RESULTS

## H.1 MORE RESULTS ON NERF SYNTHETIC DATASET

We provide the full table that contains 8 scenes of NeRF Synthetic Dataset in Table 8, as well as more qualitative results in Fig. 26.

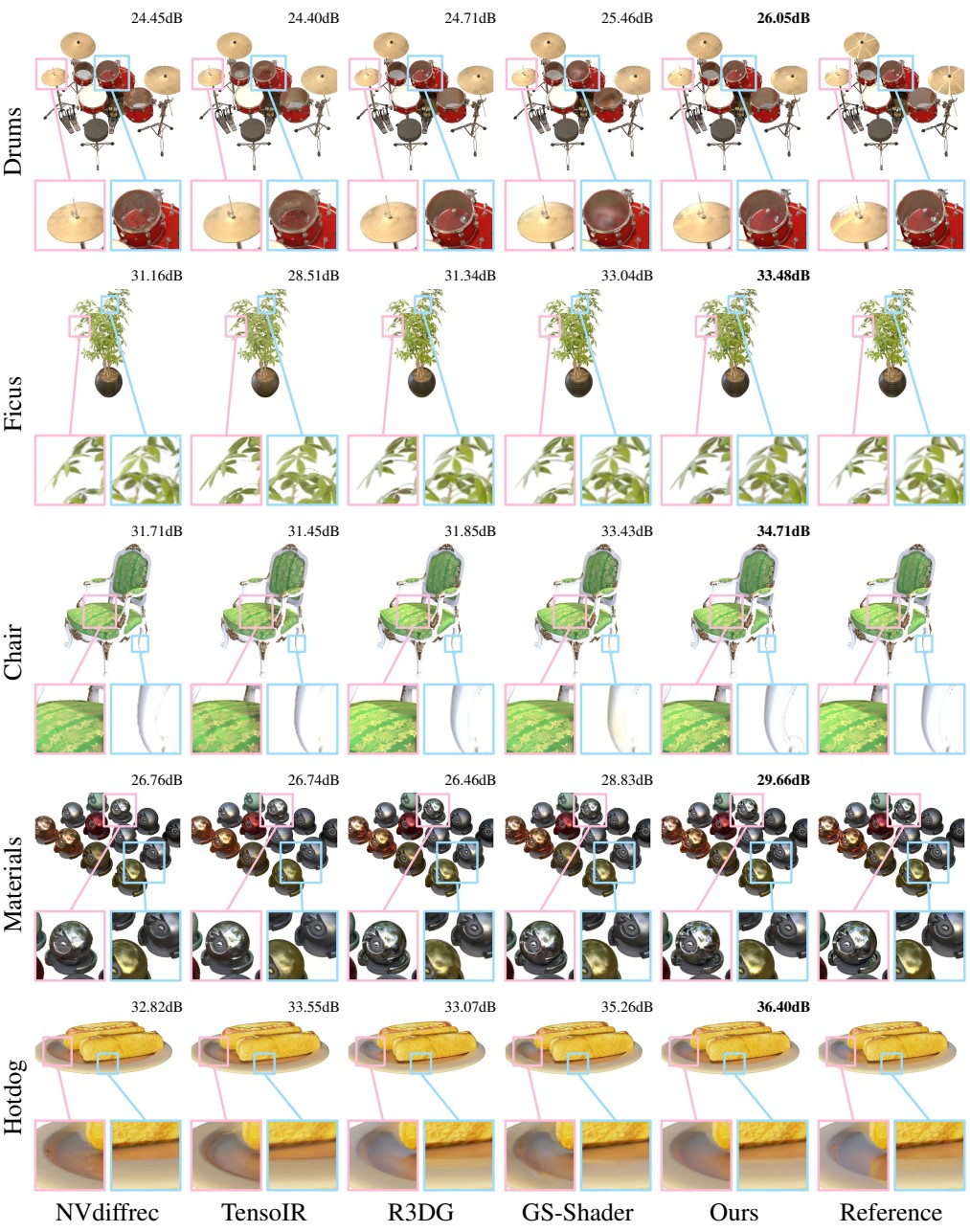

Figure 26: **More qualitative comparison of NVS on NeRF dataset**. Our method effectively recovers complex geometries, detailed textures, and non-Lambertian appearances, as shown in the sub-windows.

| Method | Relightable | Chair | Drums | Ficus | Hotdog | Lego | Materials | Mic | Ship | Avg. |
|---|---|---|---|---|---|---|---|---|---|---|
| NeRF* | No | 33.00 | 25.01 | 30.13 | 36.18 | 32.54 | 29.62 | 32.91 | 28.65 | 31.00 |
| MipNeRF* | No | 35.14 | 25.48 | 33.29 | 37.48 | **35.70** | **30.71** | 36.51 | 30.41 | 33.09 |
| 3DGS | No | **35.55** | **26.04** | **34.66** | 37.58 | 34.63 | 29.63 | **36.71** | 30.58 | **33.17** |
| TensoIR | Yes | 31.45 | 24.40 | 28.51 | 33.55 | 32.20 | 26.74 | 31.59 | 27.78 | 29.53 |
| NVdiffrec | Yes | 31.66 | 24.31 | 30.01 | 32.67 | 29.01 | 26.84 | 30.22 | 25.64 | 28.79 |
| GS-IR | Yes | 29.34 | 23.84 | 28.27 | 32.80 | 33.66 | 25.92 | 30.45 | 27.27 | 28.94 |
| R3DG | Yes | 31.85 | 24.71 | 31.34 | 33.07 | 32.69 | 26.46 | 32.74 | 28.32 | 30.15 |
| GaussianShader | Yes | 33.43 | 25.46 | 33.04 | 35.26 | 33.03 | 28.83 | 34.06 | 28.49 | 31.45 |
| Ours (stage 1) | Yes | 31.98 | 24.53 | 28.96 | 33.85 | 30.83 | 28.43 | 31.32 | 26.23 | 29.52 |
| Ours (stage 2) | Yes | **34.71** | **26.05** | **33.48** | 36.40 | 34.47 | 29.66 | 34.62 | 29.17 | 32.32 |

Table 8: Detailed quantitative NVS Comparison on NeRF dataset (PSNR↑).

## H.2 MORE RESULTS ON SYNTHESIC4RELIGHT DATASET

We provide all the other examples of Synthesic4Relight dataset in Fig. 27, Fig. 28 and Fig. 29. Detailed quantitative results are shown in Table 9.

| Scene | Novel View Synthesis | | | Relighting | | | Albedo | | | Roughness |
|---|---|---|---|---|---|---|---|---|---|---|
| | PSNR ↑ | SSIM ↑ | LPIPS ↓ | PSNR ↑ | SSIM ↑ | LPIPS ↓ | PSNR ↑ | SSIM ↑ | LPIPS ↓ | MSE ↓ |
| Air Balloons | 36.08 | 0.9806 | 0.023 | 30.89 | 0.9617 | 0.041 | 26.14 | 0.9215 | 0.068 | 0.018 |
| Chair | 40.92 | 0.9892 | 0.007 | 32.68 | 0.9795 | 0.018 | 29.59 | 0.9549 | 0.056 | 0.007 |
| Hotdog | 38.49 | 0.9876 | 0.015 | 27.54 | 0.9573 | 0.053 | 28.23 | 0.9572 | 0.087 | 0.037 |
| Jugs | 39.48 | 0.9940 | 0.007 | 35.49 | 0.9859 | 0.014 | 32.38 | 0.9732 | 0.040 | 0.005 |
| Avg. | 39.20 | 0.9881 | 0.013 | 31.65 | 0.9713 | 0.032 | 29.21 | 0.9517 | 0.063 | 0.017 |

Table 9: Detailed quantitative results of GeoSplatting on Synthetic4Relight dataset.

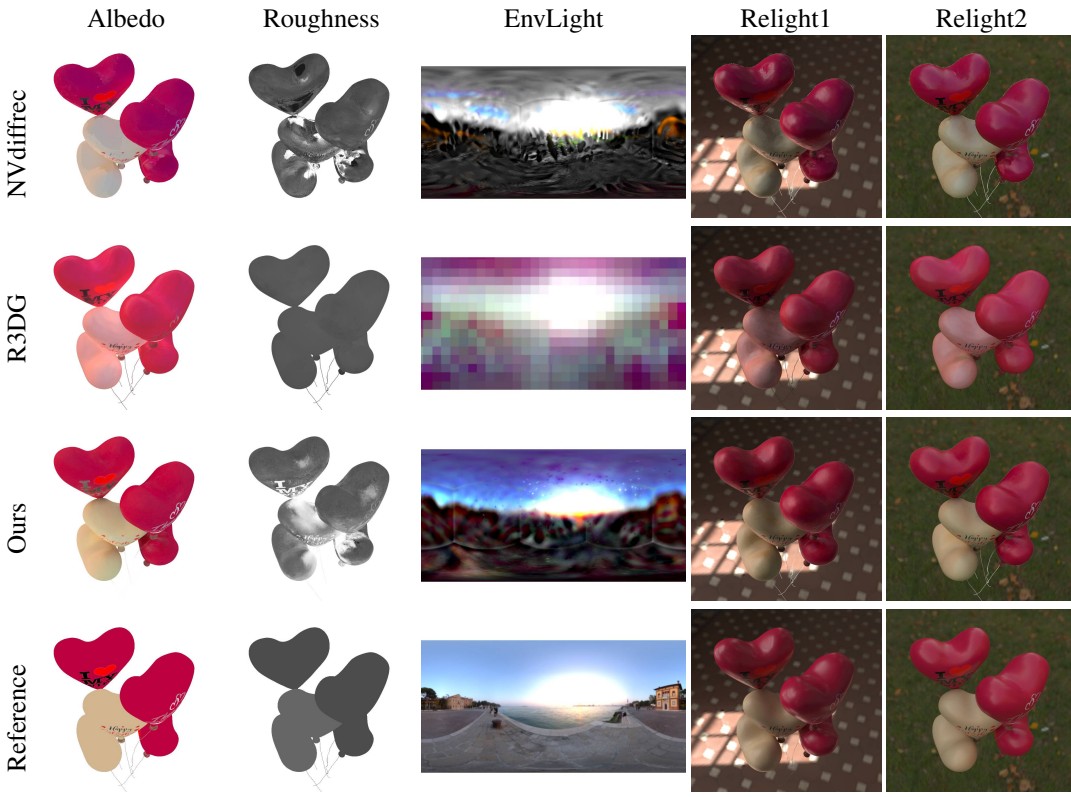

Figure 27: Qualitative comparison on Air Balloons from Synthetic4Relight dataset.

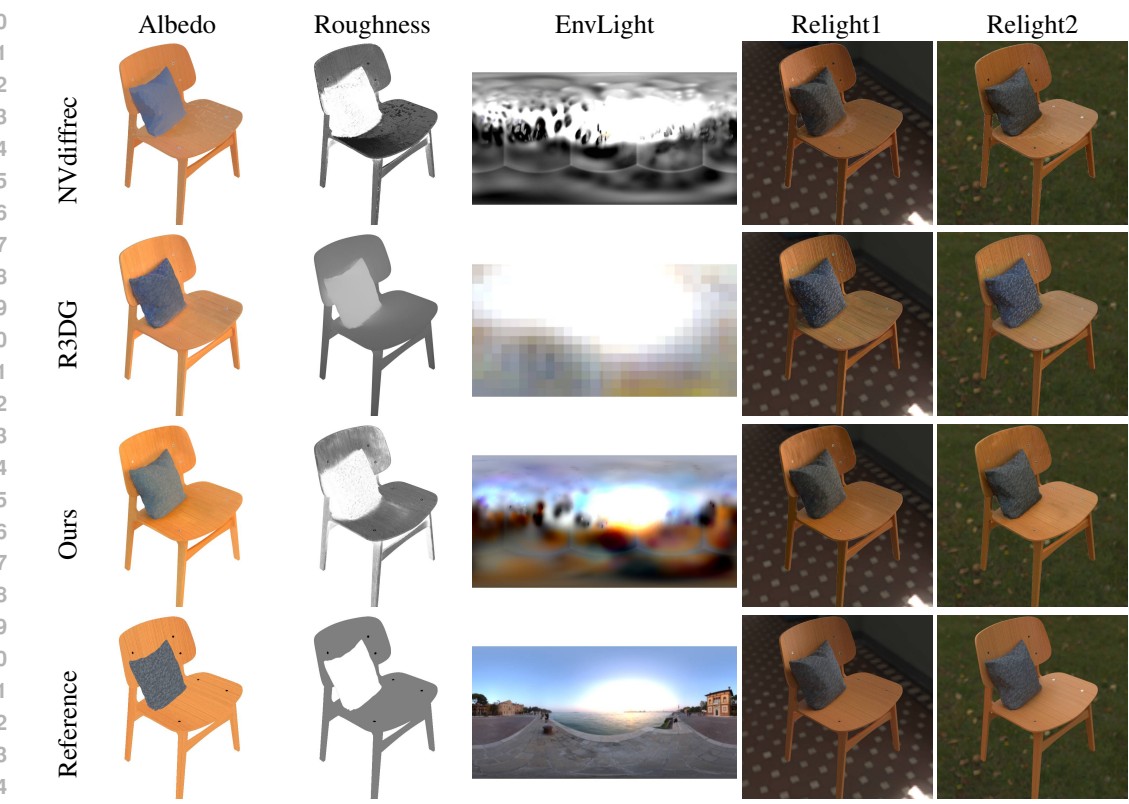

Figure 28: Qualitative comparison on Chair from Synthetic4Relight dataset.

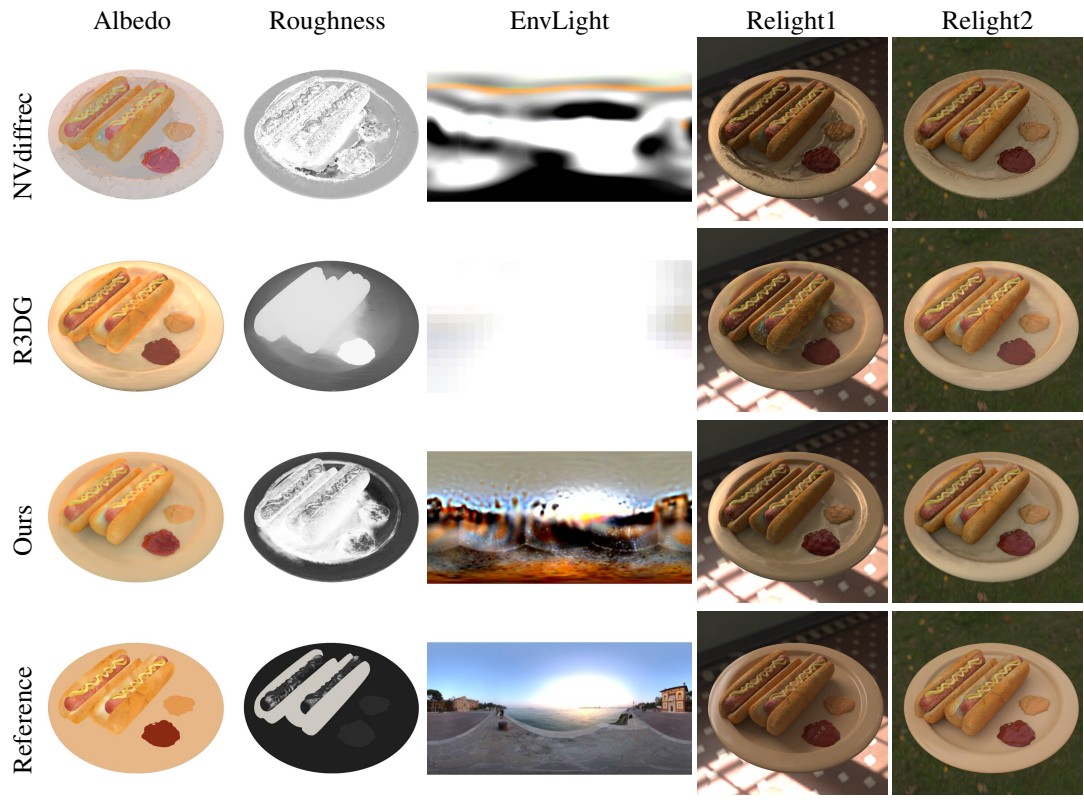

Figure 29: Qualitative comparison on Hotdog from Synthetic4Relight dataset.

## H.3 COMPARISON ON SHINY BLENDER DATASET

To demonstrate the inverse rendering capability for highly reflective cases, we conduct a comparison on the Shiny Blender dataset. The quantitative results are provided in Table 10 (for novel view synthesis) and Figure 11 (for normal quality). While several inverse rendering baselines, such as NVdiffrec, R3DG, and TenosIR, struggle with estimating the complex lighting-material interactions, resulting in degraded performance, our approach achieves the best results in both novel view synthesis and normal quality, showcasing its effectiveness in decomposing reflective objects. This is further supported by the qualitative comparison in Figures 30 and 31.

| Method | PBR | Car | Ball | Helmet | Teapot | Toaster | Coffee | Avg. |
|---|---|---|---|---|---|---|---|---|
| 3DGS* | ✗ | 27.24 | 27.69 | 28.32 | **45.68** | 20.99 | **32.32** | 30.37 |
| R3DG | SG | 27.04 | 23.81 | 26.70 | 42.61 | 20.78 | 32.05 | 28.83 |
| TensoIR | SG | 26.52 | 22.89 | 25.76 | 41.91 | 19.65 | 31.13 | 27.98 |
| NVdiffmc* | MC | 25.93 | 30.85 | 26.27 | 38.44 | 22.18 | 29.60 | 28.88 |
| NeRO* | Split-Sum+MC | 25.53 | 30.26 | 29.20 | 38.70 | 26.46 | 28.89 | 29.84 |
| NVdiffrec | Split-Sum | 24.89 | 21.76 | 27.34 | 40.94 | 23.28 | 30.62 | 28.14 |
| GS-Shader | Split-Sum | 28.41 | 29.64 | 28.46 | 42.51 | 23.22 | 31.62 | 30.64 |
| Ours | Split-Sum | **30.58** | **42.45** | **30.51** | 44.34 | **26.51** | 32.03 | **34.41** |

Table 10: **Quantitative NVS Comparison on Shiny Blender dataset (PSNR↑).** Our GeoSplatting achieves the best performance on the reflective Car, Ball, Helmet, and Toaster, whlie also significantly outperforming all baselines in terms of average PSNR. (* denotes results borrowed from the original GS-Shader paper.)

| Method | PBR | Car | Ball | Helmet | Teapot | Toaster | Coffee | Avg. |
|---|---|---|---|---|---|---|---|---|
| R3DG | SG | 4.47 | 16.50 | 5.21 | 0.64 | 9.96 | 5.48 | 7.04 |
| TensoIR | SG | 2.82 | 3.67 | 7.74 | 1.03 | 8.16 | **3.13** | 4.42 |
| NVdiffrec | Split-Sum | 9.35 | 19.91 | 9.29 | 0.88 | 10.76 | 6.08 | 9.38 |
| GS-Shader | Split-Sum | 3.68 | 5.43 | 14.59 | 2.40 | 8.69 | 7.42 | 7.03 |
| Ours | Split-Sum | **1.95** | **0.39** | **3.61** | **0.50** | **6.21** | 3.56 | **2.70** |

Table 11: **Quantitative Normal Quality Comparison on Shiny Blender dataset (MAE↓).** Our GeoSplatting achieves the best normal quality, significantly outperforming all baselines in terms of average MAE.

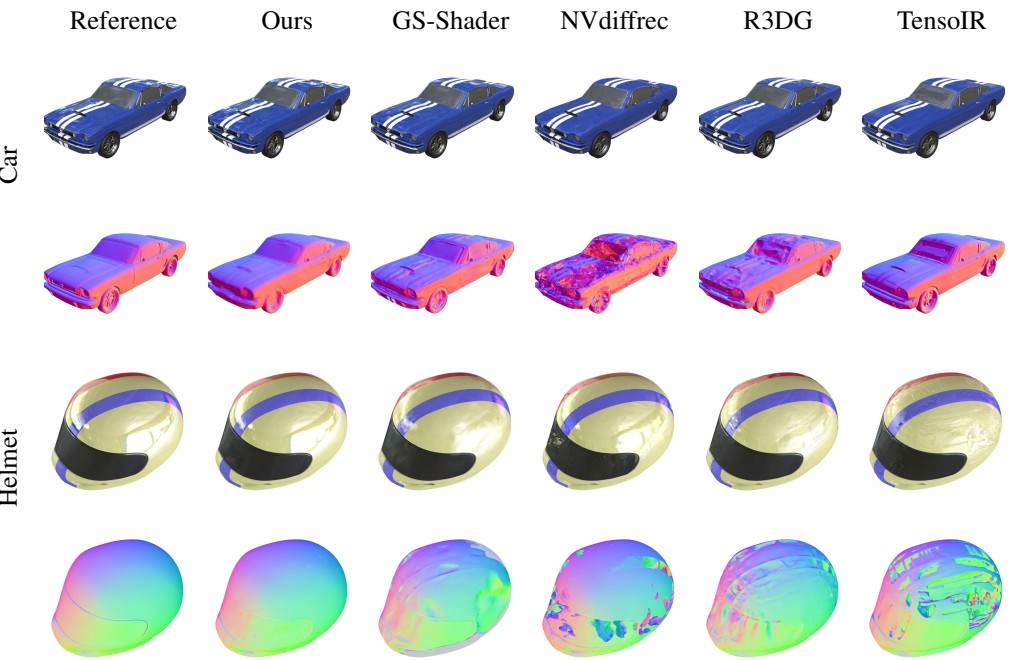

Figure 30: Qualitative Comparison on Car and Helmet from Shiny Blender dataset.

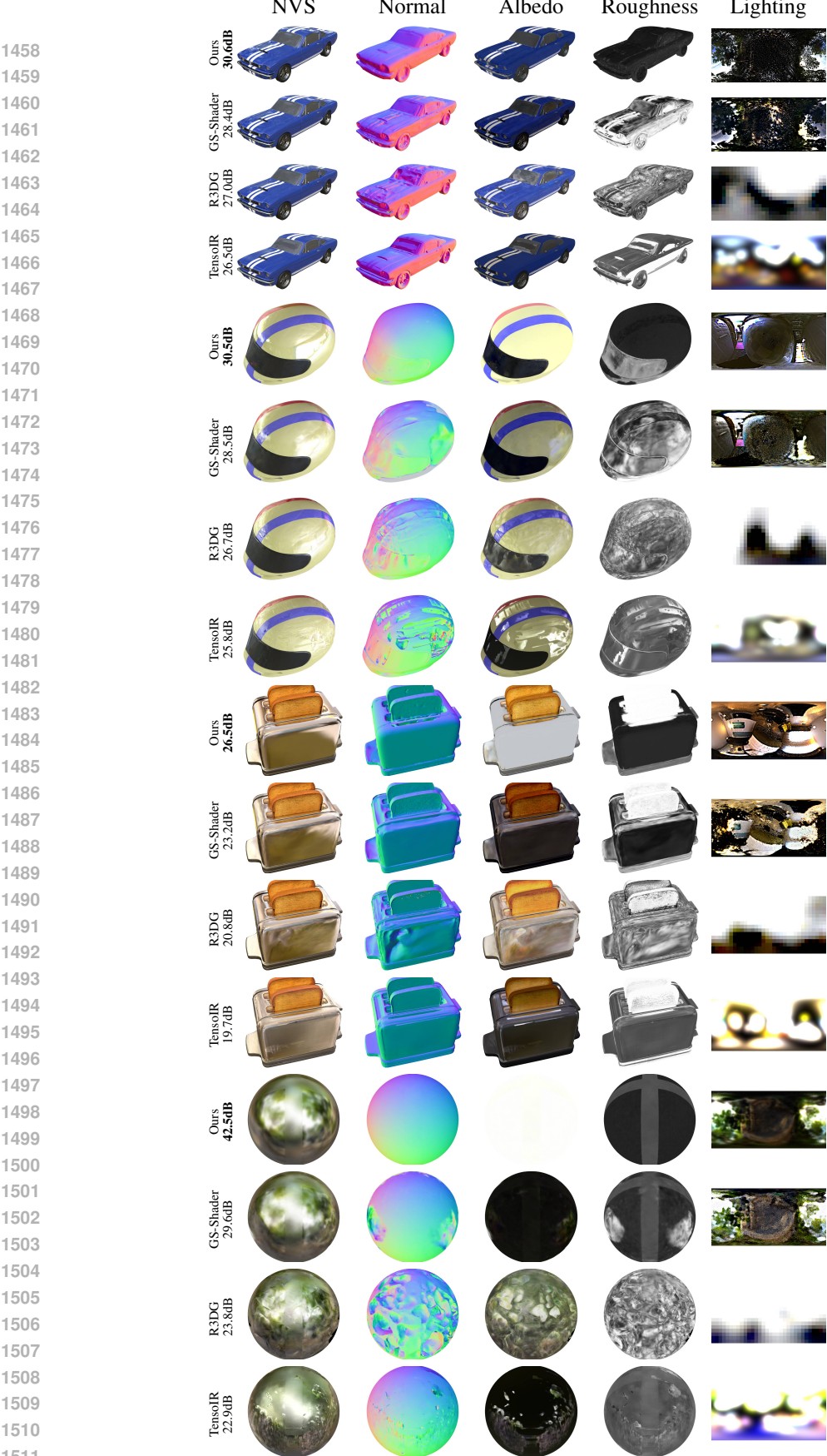

Figure 31: Qualitative Comparison of Decomposition on the Shiny Blender dataset

## H.4 COMPARISON ON TENSOIR DATASET

Additionally, We perform comparison on TensoIR dataset. Qualitative results are provided in Fig. 32. Quantitative comparison are shown in Table 12. While our method outperforms existing relightable baselines in both novel view synthesis and relighting, it also achieves comparable performance in albedo reconstruction. However, we observe a decrease in albedo reconstruction quality when transitioning from Synthetic4Relight Dataset to TensoIR Dataset. This decline is primarily due to our method partially incorporating shadows into the albedo, leading to less accurate albedo for scenes with complicated occlusion (e.g. Armadillo & Lego from TensoIR Dataset), which presents an area for improvement in future work.

| Scene | Method | Normal MAE ↓ | Novel View Synthesis PSNR ↑ | SSIM ↑ | LPIPS ↓ | Relighting PSNR ↑ | SSIM ↑ | LPIPS ↓ | Albedo PSNR ↑ | SSIM ↑ | LPIPS ↓ |
|---|---|---|---|---|---|---|---|---|---|---|---|
| Armadillo | NVdiffrec | 1.998 | 34.31 | 0.986 | 0.025 | 26.59 | 0.925 | 0.050 | 30.51 | 0.953 | 0.068 |
| | TensoIR | 1.953 | 37.92 | 0.975 | 0.042 | 34.31 | **0.975** | 0.025 | 33.11 | 0.957 | 0.057 |
| | GS-IR | 2.650 | 35.44 | 0.962 | 0.040 | 28.93 | 0.922 | 0.083 | **35.57** | 0.951 | 0.089 |
| | R3DG | 2.479 | 39.54 | 0.981 | 0.032 | 32.44 | 0.951 | 0.068 | 32.89 | 0.954 | 0.076 |
| | Ours | **1.668** | **43.68** | **0.993** | **0.005** | **34.63** | 0.970 | **0.024** | 31.58 | **0.958** | 0.042 |
| Ficus | NVdiffrec | 3.215 | 27.77 | 0.966 | 0.051 | 23.00 | 0.938 | 0.070 | 25.38 | 0.950 | 0.057 |
| | TensoIR | 4.264 | 29.78 | 0.949 | 0.037 | 24.28 | 0.946 | 0.061 | 27.74 | **0.968** | 0.030 |
| | GS-IR | 5.220 | 20.71 | 0.853 | 0.100 | 25.01 | 0.871 | 0.078 | 29.52 | 0.888 | 0.090 |
| | R3DG | 6.493 | 31.99 | 0.975 | 0.027 | **30.58** | 0.958 | 0.035 | **30.09** | 0.959 | 0.030 |
| | Ours | **2.576** | **35.45** | **0.992** | **0.006** | 30.30 | **0.978** | **0.016** | 28.12 | 0.965 | 0.026 |
| Hotdog | NVdiffrec | 5.086 | 34.85 | 0.973 | 0.044 | 23.19 | 0.910 | 0.113 | 26.65 | 0.928 | 0.117 |
| | TensoIR | 1.953 | 36.69 | 0.976 | 0.022 | **27.72** | 0.931 | 0.090 | 26.68 | 0.955 | 0.077 |
| | GS-IR | 5.145 | 31.65 | 0.961 | 0.042 | 20.40 | 0.889 | 0.112 | 21.34 | 0.907 | 0.127 |
| | R3DG | 4.865 | 33.38 | 0.972 | 0.031 | 26.64 | 0.921 | 0.091 | 26.18 | 0.951 | 0.081 |
| | Ours | **1.668** | **38.10** | **0.985** | **0.014** | 26.07 | **0.937** | 0.066 | **28.21** | **0.956** | 0.075 |
| Lego | NVdiffrec | 9.590 | 31.92 | 0.959 | 0.030 | 25.79 | 0.891 | 0.078 | 20.84 | 0.856 | 0.142 |
| | TensoIR | **5.887** | 34.95 | 0.964 | 0.020 | **27.71** | **0.926** | 0.059 | **25.86** | **0.931** | 0.072 |
| | GS-IR | 8.608 | 31.72 | 0.940 | 0.036 | 23.05 | 0.853 | 0.089 | 20.76 | 0.823 | 0.159 |
| | R3DG | 8.064 | 30.47 | 0.947 | 0.036 | 24.54 | 0.878 | 0.095 | 25.79 | 0.916 | 0.102 |
| | Ours | 7.887 | **37.07** | **0.981** | **0.011** | 27.15 | 0.920 | **0.053** | 22.06 | 0.887 | 0.113 |
| Avg. | NVdiffrec | 4.972 | 32.21 | 0.971 | 0.037 | 24.64 | 0.916 | 0.078 | 25.84 | 0.922 | 0.096 |
| | TensoIR | 4.100 | 34.84 | 0.966 | 0.030 | 28.51 | 0.945 | 0.059 | 28.35 | 0.953 | 0.059 |
| | GS-IR | 5.406 | 29.88 | 0.929 | 0.055 | 24.35 | 0.884 | 0.091 | 26.80 | 0.892 | 0.116 |
| | R3DG | 5.476 | 33.84 | 0.968 | 0.031 | 28.55 | 0.927 | 0.072 | **28.74** | 0.945 | 0.072 |
| | Ours | **3.874** | **38.57** | **0.988** | **0.009** | **29.54** | **0.951** | **0.040** | 27.49 | 0.941 | 0.064 |

Table 12: Quantitative Results on the TensoIR Dataset.

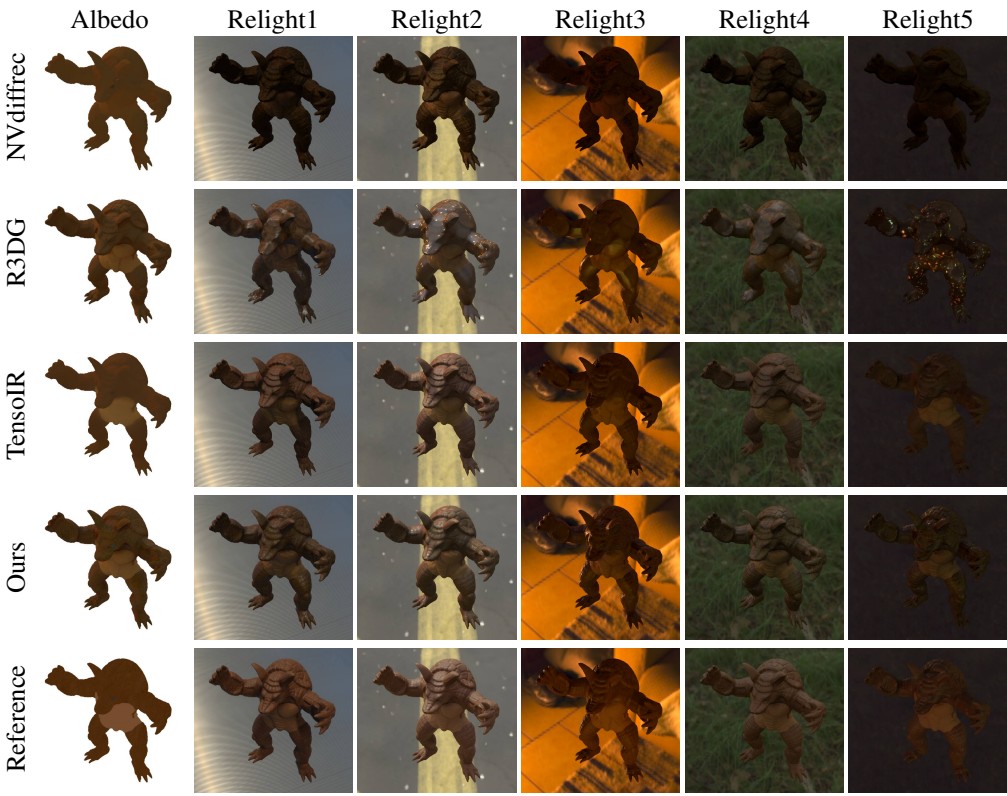

Figure 32: Qualitative comparison on Armadillo from TensoIR dataset.

## H.5    MORE RELIGHTING ON SYNTHETIC DATA

We provide more relighting results on synthetic data in Fig. 33, including Spot, Materials and Lego.

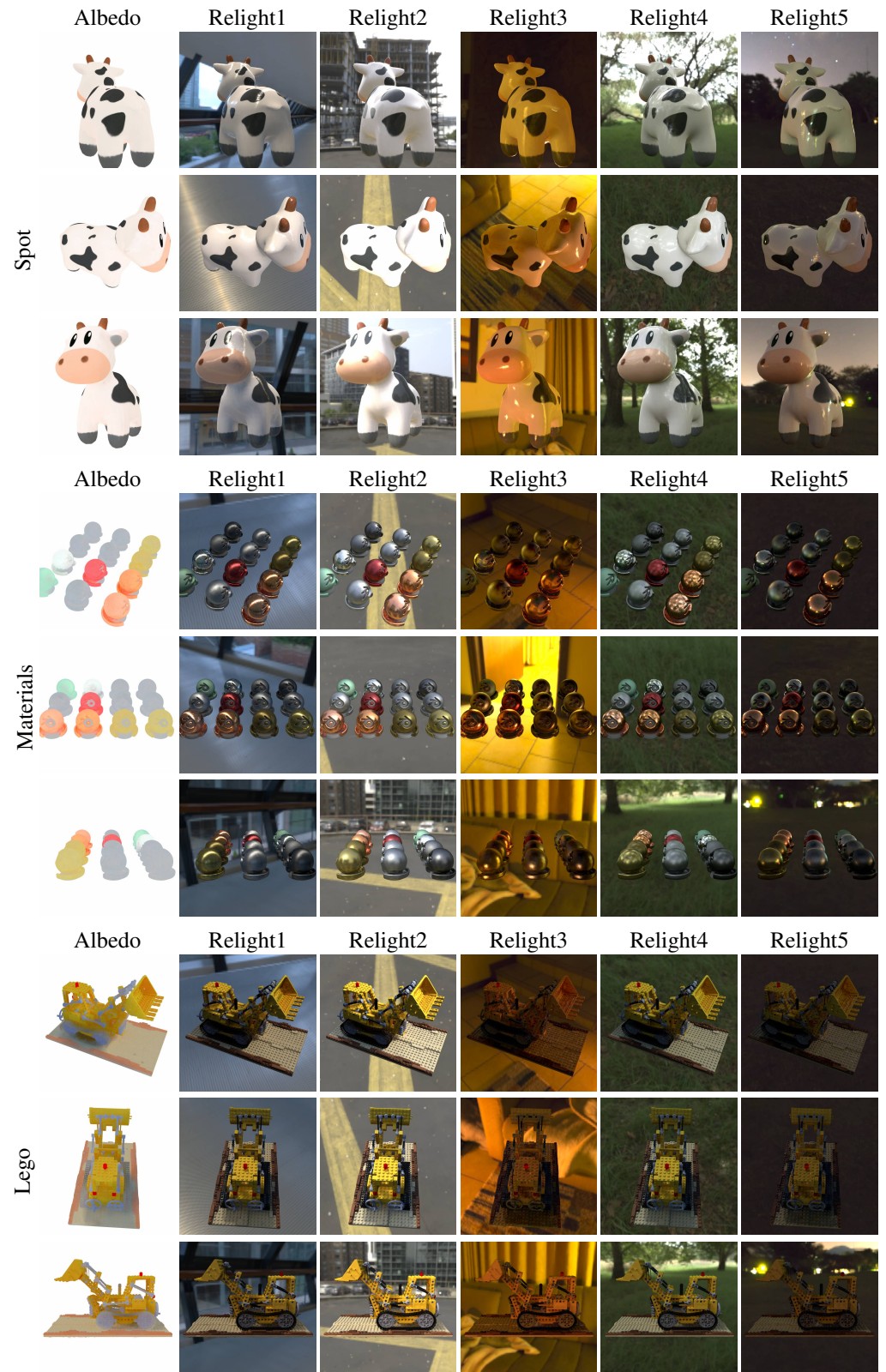

Figure 33: Relighting on synthetic data.

## H.6 RELIGHTING ON DTU DATASET

In Fig. 34, we also provide real-world relighting results from DTU Dataset.

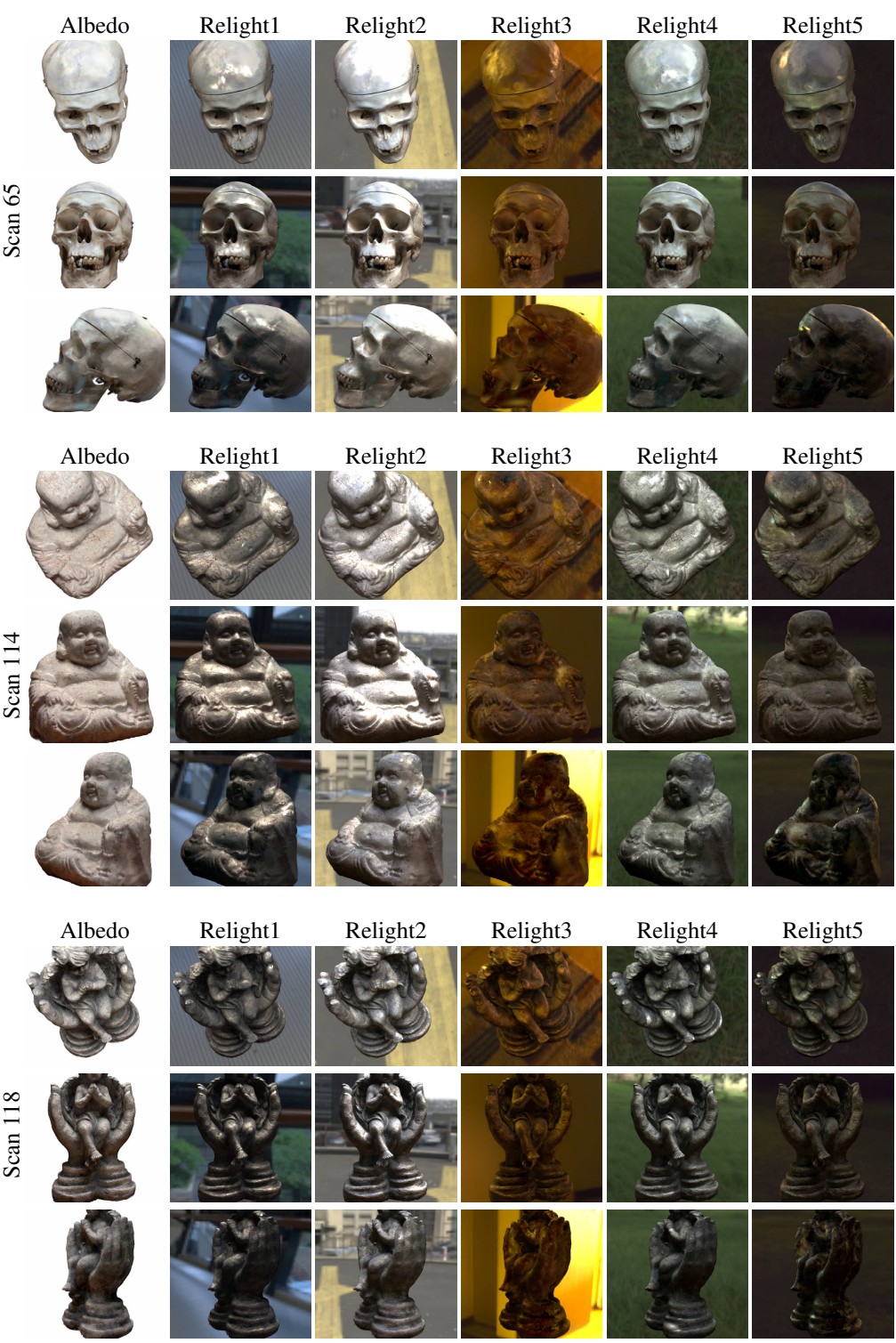

Figure 34: Relighting on DTU dataset.

