# OpenReview forum: "GeoSplating: Towards Geometry Guided Gaussian Splatling for Physically-based Inverse Rendering"
_ICLR.cc/2025/Conference — Submitted to ICLR 2025_

### Official Review · Reviewer_f2Bn · 2024-10-18

**Soundness:** 3
**Presentation:** 4
**Contribution:** 2
**Rating:** 5
**Confidence:** 5

**Summary:**

This paper is dedicated to addressing the long-standing challenges of inverse rendering using a 3DGS-based approach. The authors first introduced geometric priors to 3DGS through Flexicubes, achieving impressive geometric quality. They then adopted the standard rendering equation constraints to solve for BRDF. To further enhance the rendering quality and improve the robustness of geometry, the authors introduced residual color and a second-stage optimization that increases geometric freedom. The experimental results demonstrate that GeoSplatting achieves excellent NVS and geometric quality in both synthetic and real-world scenes, along with relatively good BRDF estimation results.

**Strengths:**

- This paper is well-written and easy to understand.
- The geometric quality and NVS quality are excellent, and introducing geometric priors and guidance through Flexicubes is novel.
- The training time is impressive, ranking among the fastest in its type of methods.

**Weaknesses:**

1. Missing reference:
   - 3DGSR: Implicit Surface Reconstruction with 3D Gaussian Splatting, Lyu et al. 3DGSR is very similar to GSDF, with the biggest difference being that the former is based on 3D-GS, while the latter is based on Scaffold-GS.
   - The related work section of this paper lacks a discussion on neural field-based inverse rendering. Before the emergence of methods based on 3D-GS, many technically innovative approaches were not adequately discussed. Examples include NeRFactor, PhySG, NVDiffrec, InvRender, NeRO and NVDiffrecMC.
2. Dataset choice on the evaluation of BRDF estimation and relighting. In lines L401-L405, the authors used the Synthetic4Relight dataset from InvRender to demonstrate the advantages of GeoSplatting. However, this dataset is not challenging because there are no shadows in the input images, masking the long-standing issues in inverse rendering related to shadow and BRDF decoupling. Using the vanilla NeRF dataset would be more convincing. For instance, in Figure 15, the hotdog scene from NeRF, which contains strong shadows, would be a better choice than the trick scene in InvRender where lighting intensity has already been reduced.
3. Questions regarding the evaluation of NVS in inverse rendering. The authors dedicated a significant portion of the paper to demonstrating the advantages of GeoSplatting in NVS (e.g., Figure 6, Table 1). This is commendable, as it highlights GeoSplatting's fidelity in reconstruction. However, I can't help but wonder—how important is NVS really in the context of inverse rendering? If NVS is truly a metric worth comparing in IR, why not use methods that don't require PBR, such as vanilla 3D-GS? In most cases, PBR constraints significantly reduce the quality of NVS. From my perspective, the most important metrics in IR are the quality of BRDF and relighting. Given that the reconstruction fidelity of base models is already sufficient today, NVS doesn't seem to be a metric that articles in the IR field should emphasize to this extent.
4. Poor decomposition results. The biggest issue with GeoSplatting is its poor decomposition quality, which is a common problem among 3D-GS-based methods. For example, in Figure 7, there is still a significant amount of shadow baked into the albedo and roughness, reflecting GeoSplatting's inability to distinguish between shadows and roughness, even though the scene in Figure 7 has minimal shadows. While ambient lighting shows improvements compared to NVDiffrec, Relightable-GS, and TensoIR, it still falls short of InvRender's level. The same problem appears in Figures 12 and 14. In contrast, Relightable-GS provides more Blender- and relighting-friendly decomposition results. Furthermore, GeoSplatting still suffers from significant hue discrepancies, which affect relighting quality. These issues are amplified in real scenes, as shown in Figure 9, where the quality of albedo and roughness becomes unacceptable (scan 65).
5. Some minor questions. In Table 1, there might be issues with the SSIM for 3D-GS and LPIPS for 2DGS. Additionally, under typical circumstances, the training time for 3D-GS at a resolution of 800x800 should not exceed 10 minutes. Since 2DGS modifies the EWA used in 3D-GS, its computational efficiency will inevitably be slower than that of 3D-GS under the same conditions, likely by around 40%. And why the training of Mip-NeRF only cost around 2.5h, which is impossible on a single RTX 4090 GPU.

**Questions:**

Are there some typo errors in the title?
- GeoSplating (-> GeoSplatting): Towards Geometry Guided Gaussian Splatling (-> Splatting) for Physically-based Inverse Rendering

---

> ### Author Response · Authors · 2024-11-20
>
> We thank the reviewer for recognizing the high efficiency of our GeoSplatting approach and the quality of our paper’s presentation. We also greatly appreciate the detailed suggestions provided. Below, we address the concerns point by point. For complex issues, please refer to the appendix in our revised submission for a more detailed explanation, if necessary.
>
> ## 1. Inadequate Discussion
>
> We thank the reviewer for listing many valuable related works. For the carefully suggested works, we will provide a brief discussion as follows and will reference them in our paper.
>
> - **3DGSR.**
>
>   It maintains the **stochastic nature** of 3DGS geometry, using the SDF-to-opacity transformation to align Gaussians with a deterministic surface defined by the SDF field. During training, the stochastic geometry converges to this surface, achieving excellent geometry reconstruction.
>
>   GeoSplatting, on the other hand, prioritizes precise modeling of light-geometry interactions. It uses an explicit isosurface to represent **deterministic geometry**, enabling accurate normal modeling for complex lighting effects. Additionally, it extends 3DGS with PBR attributes for efficient, high-quality differentiable rendering, offering robust inverse rendering capabilities, especially for specular scenes.
>
> - **Isosurface-based inverse rendering.**
>
>   Isosurface-based methods (e.g. NVdiffrec, NVdiffrecmc) employ differential mesh-based rendering processes, which are prone to falling into local minima due to the deterministic visibility of isosurfaces.
>
>   In contrast, our GeoSplatting employs Gaussian splatting as a more powerful differentiable PBR rendering technique.
>
> - **Neural-field-based inverse rendering.**
>
>   Several neural-field-based methods have been proposed for inverse rendering, integrating neural fields with PBR rendering, such as NeRFactor, PhySG, InvRender, TensoIR, and NeRO.
>
>   While these methods offer varied solutions, they are much less efficient than 3DGS-based approaches in FPS and training time, and their rendering quality, lighting modeling, and relighting capabilities are limited by the sampling-intensive nature of neural fields.
>
>   In contrast, our GeoSplatting is efficient and is able to model high-frequency light-material interaction by integrating 3DGS with isosurfaces.
>
> ## 2. Dataset Choice
>
> We appreciate the reviewer’s suggestion that the Synthetic4Relight dataset may not be challenging enough to evaluate shadow modeling. However, we would like to respectfully point out that the suggested *Hotdog* scene from the NeRF Synthetic dataset can be too challenging for most of the existing inverse rendering methods to achieve effective material-shadow decoupling, as shown in Fig. 18 (Appendix D.2).
>
> Therefore, to follow the reviewer’s constructive suggestion and to provide more results to evaluate shadow and BRDF decoupling, we performed additional experiments using the TensoIR Synthetic dataset. Results are shown in Fig. 32 (Appendix H.4), with quantitative data provided in Table 12 (Appendix H.4), which demonstrate GeoSplatting’s best relighting performance and its comparable albedo quality to state-of-the-art baselines including the 3DGS-based R3DG and the neural-field-based TensoIR.

---

> ### Author Response · Authors · 2024-11-20
> **(Continued)**
>
> ## 3. NVS Comparison
>
> Thanks for the reviewer’s constructive concerns that novel view synthesis (NVS) quality may not be critical for inverse rendering. This issue is very valuable and worth further discussion. **We would respectfully argue that the importance of NVS quality in the context of inverse rendering depends on the dataset, and will carefully clarify as follows:**
>
> **NVS quality is not important for diffuse scenes** (in Fig. 6): NVS quality is unable to reflect ability in modeling complex light interactions, such as inter-reflections and shadows.
>
> **NVS quality is important for highly specular scenes** (e.g., *Materials*): NVS quality provides strong evidence of the ability to model high-frequency lighting-material effects, like those on shiny surfaces. Additionally, many works, including Ref-NeRF, GaussianShader, and NeRO, use NVS experiments on the Shiny Blender dataset, where the scenes are highly specular, to demonstrate their ability to model high-frequency lighting-material effects.
>
> As a result, while Table 1 and Fig. 6 offer limited insights into inverse rendering performance, we also conducted NVS comparisons on the Shiny Blender dataset that features highly specular scenes, following Ref-NeRF and GaussianShader, to validate our ability to model high-frequency lighting-material effects. Please see Appendix H.3 for more details.
>
> ## 4. Decomposition Quality
>
> The reviewer's concern about GeoSplatting's limitations in shadow modeling is very valuable in the context of inverse rendering. However, while shadow and inter-reflection effects are important in inverse rendering, we respectfully highlight that modeling high-frequency lighting-material effects is equally crucial. Given the **“impossible trinity”** explained in our general response, we would also clarify that none of the existing works can address reflective surface, shadow, and efficiency at the same time.
>
> **In this context, we claim our GeoSplatting achieves state-of-the-art inverse rendering performance for the following reasons: (1)** For specular objects, GeoSplatting surpasses all baselines in rendering and normal quality, delivering superior decomposition results, which is evidenced by experiments in Appendix H.3. **(2)** For general inverse rendering tasks including occluded cases, GeoSplatting leads in rendering quality, training efficiency, relighting performance, and normal quality, with decomposition results on par with the state-of-the-art methods, which is evidenced by Table 2 and additional experiments in Appendix H.4.
>
> **Additionally, we highlight that GeoSplatting can model shadow effects by introducing a learnable occlusion term (Appendix D.2) in the PBR pipeline.** The qualitative results show that the occlusion term brings significant improvements in material-shadow decoupling, compared to those in the previously submitted version, as shown in Fig. 28-29 of the revised submission (e.g., *Hotdog, Chair* from Synthetic4Relight), with quantitative evidence provided in Table 2 (Sec. 4.2).
>
> Again, we thank the reviewer for the insightful and constructive suggestions of the reviewer and agree that good work should focus on the long-standing issues in the field, to contribute to the community. We provide more discussions in Appendix D, sincerely hoping to find common ground with the reviewer on the advanced aspects of inverse rendering, while kindly requesting reconsideration of GeoSplatting's strengths and weaknesses.
>
> ## 5. Baseline Metrics and Training Times
>
> Thanks for the kind suggestions. We apologize for the discrepancies in the reported metrics.
>
> 1. **3DGS.** The training time for 3D-GS is based on our reimplementation using the gsplat repo on Github, which impacts performance. After using the original codebase, we confirmed a training time of approximately 5.4 minutes, with no errors in the SSIM metric upon re-evaluation.
> 2. **2DGS.** The 2D-GS metrics are directly from the original codebase, so we believe there is no error.
> 3. **Mip-NeRF.** The reported 2.5 hours was cited from the original paper, but we missed that its experiments are conducted on TPU. On a single RTX 4090 GPU, our reimplementation shows a training time of ~48 hours (1152 minutes). We’ve updated these metrics in the revised version.
>
> ## Q1. Typos in the Title
>
> As the reviewer carefully pointed out, there are typos in the title and we’ve fixed them in our revised submission.

---

> > ### Comment · Reviewer_f2Bn · 2024-11-23
> >
> > I greatly appreciate the authors for providing such a detailed explanation in response to my questions. Overall, I agree with the authors' viewpoints. Here are some minor suggestions I would like to offer:
> >
> > - I believe that the `Synthetic4Relight` dataset used by InvRender is essentially the same as the `TensoIR` dataset, which features diffuse scenes with light intensity so weak that no shadows are present. Therefore, using the TensoIR dataset does not seem convincing.
> >
> > - Considering the unparalleled training speed of Geo-Splatting, I think Geo-Splatting could focus on training speed and reflective scenes while de-emphasizing the decomposition results for general scenes. In inverse rendering, Geo-Splatting does not have an advantage in decomposition quality for general scenes compared to previous methods (the more mature frameworks currently import albedo, roughness, metallic, etc., directly into Blender for relighting). The current Geo-Splatting provides a new representation method, which will undoubtedly inspire new ideas in the community. However, an excessive emphasis on achieving SOTA decomposition quality for general scenes may distort the community's understanding of this method.
> >
> > Therefore, I suggest that the authors highlight the training speed, geometric quality, and the improvements in reflective scenes brought by geometric priors. At the same time, the poorer decomposition quality in general scenes and in scenes with strong shadows could be objectively presented as failure cases to the readers. The innovation in representation and the significant improvement in training speed are already sufficient to make Geo-Splatting a good paper.

---

> > > ### Author Response · Authors · 2024-11-23
> > >
> > > Thanks again for the reviewer's constructive suggestions. We will address the concerns as follows:
> > >
> > > ---
> > >
> > > ### TensoIR Dataset
> > >
> > > We acknowledge that the light intensity in scenes from the TensoIR dataset is not as strong as that in the Hotdog scene from NeRF Synthetic. While none of the compared baselines can adequately handle the strong shadow effects present in the Hotdog scene (as illustrated in Appendix D.2), we believe that the shadow effects in the Armadillo scene are sufficient to highlight the performance gap in shadow modeling between GeoSplatting and SG-based methods such as TensoIR. As the reviewer noted, we discussed Figure 32 in General Response section, which illustrates how our relighting performance is affected by shadow effects.
> > >
> > > ---
> > >
> > > ### Contribution Claims
> > >
> > > We greatly appreciate the valuable suggestions of the reviewer and have accordingly de-emphasized the decomposition performance for general scenes. In the General Response section, we have provided a revised contribution statement, which we restate here:
> > >
> > > We propose GeoSplatting, a novel inverse rendering method that leverages a hybrid representation of 3DGS and isosurfaces. By inheriting the excellent efficiency of 3DGS and incorporating accurate isosurface geometry guidance, GeoSplatting achieves state-of-the-art inverse rendering performance in reflective scenes, which demonstrates its impressive ability in modeling high-frequency light-material interaction. Additionally, for general cases, GeoSplatting delivers decomposition results that are comparable to those of state-of-the-art inverse rendering baselines. Extensive quantitative and qualitative results are provided to support our claims.
> > >
> > > ---
> > >
> > > ### Thanks
> > >
> > > Once again, we sincerely thank the reviewer for the valuable discussions, which have provided us with a deeper understanding of the field of inverse rendering, particularly in the areas of shadow modeling and neural-field-based methods. We would greatly appreciate it if the reviewer could consider raising the rating of our paper, provided that all concerns have been well addressed.

---

> > > ### Author Response · Authors · 2024-11-26
> > >
> > > We sincerely thank the reviewer for the kind reconsideration and for raising the rating. The reviewer's constructive suggestions have greatly enhanced our understanding, and we hope our discussion has been equally valuable to the reviewer.

---

> > > ### Author Response · Authors · 2024-11-28
> > > **Request for Further Feedback during Extended Rebuttal Phase**
> > >
> > > ### **Dear Reviewer f2Bn**,
> > >
> > > We sincerely invite you to provide further feedback on the contributions of our work. While we greatly appreciate your comment that our restated contributions are "good enough," we also follow your valuable suggestions to further validate GeoSplatting's performance on reflective scenes, especially compared to state-of-the-art neural-field-based inverse rendering approaches, such as NeRO.
> > >
> > > Therefore, we provide quantitative experiments comparing NeRO. We would like to highlight that our GeoSplating method achieves the best PSNR and normal quality (MAE) among all NeRF-based and 3DGS-based inverse rendering methods.
> > >
> > > ---
> > >
> > > | Method    | Training Time | Normal MAE (Diffuse Cases) | Normal MAE (Specular Cases) | PSNR (Specular Cases) |
> > > | --------- | ------------- | -------------------------- | --------------------------- | --------------------- |
> > > | NeRO      | 600min        | 4.97                       | 2.36                        | 29.84                 |
> > > | TensoIR   | 270min        | 4.10                       | 4.42                        | 27.98                 |
> > > | NVdiffrec | 72min         | 4.97                       | 9.38                        | 28.14                 |
> > > | R3DG      | 140min        | 5.48                       | 5.48                        | 28.83                 |
> > > | GS-Shader | 63min         | 6.53                       | 7.03                        | 30.64                 |
> > > | Ours      | **20min**     | **3.87**                   | **2.28**                    | **35.05**             |
> > >
> > > *Note that the "diffuse cases" denote scenes from the TensoIR Synthetic dataset and the "specular cases" denote scenes from the Shiny Blender dataset.*
> > >
> > > ---
> > >
> > > In light of these results, we sincerely hope that you could reconsider our contributions, as we believe these results establish our approach as a state-of-the-art inverse rendering method for reflective scenes, addressing your earlier concern regarding potential over-claims. We will also release all the code to facilitate future research. We are more than happy to address any additional concern you may have. Your feedback would be invaluable in helping us address any remaining concerns.
> > >
> > > Best regards,
> > >
> > > Authors

---

> > > > ### Author Response · Authors · 2024-12-02
> > > >
> > > > Dear Reviewer f2Bn,
> > > >
> > > > We sincerely thank you for your valuable feedback, time, and effort during the rebuttal period. We greatly appreciate the insightful discussion, which has strengthened our paper and contributed to score improvements. Following your suggestions, we have added experiments to demonstrate our method's efficiency and high performance in reflective scenes. We hope our responses have fully addressed your concerns. If there are any remaining issues, we are happy to provide further clarifications or additional experiments as needed. We would greatly appreciate it if you could consider raising the rating of our paper.
> > > >
> > > > Thank you!
> > > >
> > > > Best regards,
> > > > The Authors

---

> > > > > ### Comment · Reviewer_f2Bn · 2024-12-02
> > > > >
> > > > > Many thanks to the authors for their efforts. I have great respect for the work the authors put in during the rebuttal and discussion period, which is also the reason I increased my score from 3 to 5.
> > > > >
> > > > > However, due to the poor decomposition results, I will maintain my current score.

---

> ### Author Response · Authors · 2024-12-02
>
> We sincerely thank the reviewer for the invaluable discussion, time, and effort. As mentioned, this discussion has certainly strengthened our paper. While the reviewer may think the decomposition results are poor, we respectfully clarify that these results are quantitatively competitive with TensoIR and R3DG, and the claims should be confined to the context of diffuse cases. Following the reviewer's suggestion, We've already de-emphasized decomposition performance for diffuse objects.
>
> Additionally, we would like to highlight that our current contributions focus on:
> 1. Achieving the best efficiency (Table 1),
> 2. Providing the highest normal quality (Table 1 & 12),
> 3. Achieving the best specular surface decomposition performance (Table 1), and
> 4. Delivering competitive diffuse surface decomposition performance (Table 2 & Table 12).
>
> We believe that (1), (2), and (3) are equally as important as (4) in the context of inverse rendering, which makes GeoSplatting ”a good paper”, as praised by the reviewer. We fully acknowledge that decomposition is a critical aspect and understand that this might be the primary concern of the reviewer. However, we respectfully suggest that a single point of evaluation should not be the sole basis for rejecting the paper. Once again, we sincerely thank the reviewer for the thoughtful feedback!

---

### Official Review · Reviewer_DXwn · 2024-10-28

**Soundness:** 2
**Presentation:** 2
**Contribution:** 2
**Rating:** 5
**Confidence:** 4

**Summary:**

The paper introduce GeoSplatting, a novel hybrid representation that augments 3DGS with explicit geometric guidance and differentiable PBR equations.
Contributions:
1. extract isosurface mesh from a flexicube, then sample 3D gaussian points on triangle faces.
2. apply PBR equations in a deferred rendering manner.

**Strengths:**

1.The paper presents a novel pipeline which extract issosurface from flexicube and then convert to 3DGS for rendering.
 2. The experiment results show good performance in various datasets.

**Weaknesses:**

1. Many typos.
- in the title "geosplating", "splatling"
- equation 2: ","
- line254-255: "rasterizationto"
- Figure14(ours,Relit2)
2. lack of novelty: the physically-based rendering after rastization, sample 3D gaussian on triangle face, shift gaussian position on surface, these are all proposed by previous methods
3. I think the geometry compare in Table3 in unfair, as GeoSplatting only achieves 1 out of 4 best quality. And the only best scene (spot) is too simple, which can be represent by the 96^3 grid, but for a complex scene, such as the lego in Figure10, it's obvious that the geometry is not good at all.

**Questions:**

1. how to compute the normal map in stage2 (Figure10)

---

> ### Author Response · Authors · 2024-11-20
>
> We sincerely thank the reviewer for the constructive suggestions and valuable reviews. We will address the concerns point by point in the responses below. For complicated issues, please refer to the appendix in our revised submission for a detailed explanation, if necessary.
>
> ## 1. Typos
>
> Thanks for the kind suggestions. In our revised submission, we’ve fixed most typos.
>
> ## 2. Novelty
>
> We greatly thank the reviewer for the constructive concern about the novelty of our work, but would respectfully hold a different opinion from the reviewer. We would like to emphasize our novelty as follows:
>
> - **Our MGAdapter is a novel Mesh-to-Gaussian module that explores the key problems in mesh-Gaussian alignment, including (1)** The shape of the Gaussians does not align well with the triangle mesh in tangential directions, particularly near edges. **(2)** Strict alignment in normal directions reduces appearance quality (see Appendix B.4). **(3)** Standard 3DGS uses a fixed number of Gaussian points, but the Mesh-to-Gaussian module needs optimization for a variable number of points.
>
>   Our MGAdapter effectively addresses all of these challenges, providing accurate geometric guidance while preserving the rendering quality as well as the efficiency of the standard 3DGS.
>
> - **Our integration of isosurface geometry guidance with Gaussian-based PBR rendering is a novel methodology in the field of inverse rendering**, which bridges isosurface and Gaussian together, naturally resulting in more accurate local geometry and normal directions, therefore bringing better inverse rendering performance.
>
>   Extensive experiments demonstrate the effectiveness of such an integration. Specifically, GeoSplatting achieves **the best relighting performance, best normal quality, and state-of-the-art decomposition performance** on multiple datasets including Synthetic4Relight, TensoIR Synthetic, and Shiny Blender.

---

> ### Author Response · Authors · 2024-11-20
> **(Continued)**
>
> ## 3.1 Geometry Quality Comparison in Table 3
>
> We appreciate the reviewer’s valuable concerns about the geometry quality of GeoSplatting and would like to clarify about geometry quality of GeoSplatting. Specifically,
>
> - **Our comparison in Table 3 is not “unfair”, as we only claim comparable performance in geometry reconstruction quality, not the best.**
>
>   The original statement, "Overall, our approach achieves the best average geometry performance" (L454-L455), is misleading. We do not claim superiority in geometry reconstruction quality, as GeoSplatting is designed for inverse rendering tasks.
>
>   Instead, the results in Table 3 show that our method achieves comparable geometry recovery performance to state-of-the-art surface reconstruction techniques, demonstrating the effectiveness of our intermediate mesh guidance strategy.
>
>   Considering the reviewer’s valuable concern, we remove the misleading Avg. column of Table 3 in our revised submission for a clearer representation.
>
> - **To further support our claim about normal quality, we conduct additional experiments to demonstrate GeoSplatting’s accurate normal estimation: (1)** On TensoIR Dataset: Our method achieves the best average normal quality (mean angle error, MAE) among relightable baselines (see Table 12 in Appendix H.4, Line 1521-1540). **(2)** On Shiny Blender Dataset: Our approach significantly outperforms all listed inverse rendering methods in MAE (see Table 11 in Appendix H.3, Line 1426-1435).
>
> ## 3.2 Spot
>
> **While the best-performing scene (Spot) features relatively simple geometry, we respectfully disagree with the reviewer’s characterization of it as "too simple.”** We hope the reviewer provides more context of the meaning of “simple,” and will carefully provide further explanation as follows:
>
> - **If "simple" means simple to reconstruct,** we would like to highlight that the Spot scene remains challenging for reconstruction. This is evidenced by the failure of several powerful geometry reconstruction baselines, such as 2DGS and NeuS2. It would be contradictory to describe the case as "simple" when the leading reconstruction methods struggle with it. 2DGS and NeuS2 fail to capture specular surfaces in Spot, mainly due to their inability to model complicated light-material interactions.
> - **If "simple" means the Spot has a simple geometry and lacks challenge,** we would like to emphasize that our discussion is in the context of inverse rendering, not purely geometric reconstruction. We want to emphasize that GeoSplatting is designed to address inverse rendering challenges, with a particular focus on capturing complex material-light interactions. For specular surfaces in Spot, GeoSplatting successfully models complicated light-material interaction, leading to state-of-the-art geometry quality in this case.
>
> ## 3.3 Lego
>
> The reviewer concerns that GeoSplatting’s geometry on the Lego scene is “not good at all”. However, we would respectfully disagree with it and would like to provide convincing evidence. Therefore, we include a comparison of normal quality on the TensoIR dataset (Appendix H.4 in our revised submission), where GeoSplatting on the Lego scene achieves comparable normal quality to the best method, TenosIR, despite its $300^3$ grids over our $96^3$ ones.
>
> Additionally, although the geometry quality of the Lego scene is constrained by the resolution of the FlexiCubes, we examine the effect of FlexiCubes resolution on GeoSplatting's performance in Appendix E. While higher-resolution FlexiCubes provide more precise normal guidance, the associated increase in computational cost and training time may outweigh the benefits.
>
> ## Q1. Normal Map
>
> Since each Gaussian point has a normal attribute, the normal map can be generated by directly splatting the normal attributes onto the screen, similar to how RGB colors are rendered. Most 3DGS-based inverse rendering methods produce normal maps in this way.

---

> > ### Author Response · Authors · 2024-11-25
> >
> > Dear Reviewer DXwn,
> >
> > We greatly appreciate the reviewer's time for review. We sincerely hope that the reviewer's concerns about typos, novelty, and geometry quality are well addressed, and we are willing to provide detailed explanations if there are any further concerns raised by the reviewer. We would appreciate it if the reviewer could provide more feedback and kindly reconsider the contribution of our work. Thank you again!
> >
> > Best wishes,
> >
> > Authors

---

> ### Author Response · Authors · 2024-11-28
> **Request for Further Feedback during Extended Rebuttal Phase**
>
> ### **Dear Reviewer DXwn,**
>
> We greatly thank for your time and effort in the rebuttal period, and would greatly appreciate it if you could provide feedback on our replies. We would like to discuss further if there are any remaining concerns.
>
> We would like to provide a brief quantitative comparison here, to address your earlier concern regarding potentially inadequent experimental results. We would like to highlight that our GeoSplating method achieves the best PSNR and normal quality (MAE) among all NeRF-based and 3DGS-based inverse rendering methods, demonstrating our approach as a state-of-the-art inverse rendering method for reflective scenes and validating GeoSplatting's excellent ability in capturing high-frequency lighting-material interactions.
>
> ---
>
> | Method    | Training Time | Normal MAE (Diffuse Cases) | PSNR (Diffuse Cases) | Normal MAE (Specular Cases) | PSNR (Specular Cases) |
> | --------- | ------------- | -------------------------- | -------------------- | --------------------------- | --------------------- |
> | NeRO      | 600min        | 4.97                       | 29.08                | 2.36                        | 29.84                 |
> | TensoIR   | 270min        | 4.10                       | 34.84                | 4.42                        | 27.98                 |
> | NVdiffrec | 72min         | 4.97                       | 32.21                | 9.38                        | 28.14                 |
> | R3DG      | 140min        | 5.48                       | 33.84                | 5.48                        | 28.83                 |
> | GS-Shader | 63min         | 6.53                       | 37.54                | 7.03                        | 30.64                 |
> | Ours      | **20min**     | **3.87**                   | **38.57**            | **2.28**                    | **35.05**             |
>
> *Note that the "diffuse cases" denote scenes from the TensoIR Synthetic dataset and the "specular cases" denote scenes from the Shiny Blender dataset.*
>
> ---
>
> In light of these results, we sincerely hope that you could reconsider our contributions as we believe that GeoSplatting proposes a novel framwork, employing geometry guidance to both efficienctly and effectively enhance inverse rendering ability, which we see valuable to the community. We will also release all the code to facilitate future research. We are more than happy to address any additional feedback you may have.
>
> Best regards,
>
> Authors

---

> > ### Comment · Reviewer_DXwn · 2024-11-30
> >
> > I appreciate the authors' detailed response, which has addressed most of my concerns. I also agree with Reviewer f2Bn's suggestion that the authors should focus more on reflective scenes, as Geo-Splatting shows superior qualitative and quantitative results with reflective objects compared to previous methods. However, for datasets like TensoIR and Synthetic4Relight, Geo-Splatting's performance in normal quality is less competitive, and it does not account for visibility modeling, which limits its performance in relighting tasks. Additionally, Geo-Splatting still requires a second stage using the vanilla 3DGS, which undermines the effectiveness of its differentiable isosurface extraction method. Moreover, methods like 3DGS-DR can handle reflective objects well and could be used for geometry initialization instead of FlexiCube, as the latter is limited by its resolution. As a result, I maintain my original score.

---

> > > ### Author Response · Authors · 2024-12-02
> > >
> > > Dear Reviewer DXwn,
> > >
> > > We sincerely appreciate your valuable feedback and thank you for your time and effort during the rebuttal period. We greatly appreciate the insightful discussion and have included further explanations to show the effectiveness of GeoSplatting. We hope our responses have fully addressed your concerns. If there are any remaining issues, we are happy to provide further clarifications or additional experiments as needed. We would greatly appreciate it if you could consider raising the rating of our paper.
> > >
> > > Thank you!
> > >
> > > Best regards,
> > > The Authors

---

> ### Author Response · Authors · 2024-11-30
>
> ## **Dear Reviewer DXWn,**
>
> We sincerely thank the reviewer for the time and effort in evaluating our work.  We are happy that most concerns have been addressed.  The reviewer raises new concerns as listed:
>
> 1. "... agree with Reviewer f2Bn ... should focus more on reflective scenes."
> 2. "... for datasets like TensoIR and Synthetic4Relight, Geo-Splatting's performance in normal quality is less competitive, and it does not account for visibility modeling..."
> 3. "... Geo-Splatting still requires a second stage ... which undermines the effectiveness of its differentiable isosurface extraction method"
> 4. "... methods like 3DGS-DR can handle reflective objects well ... could be used for geometry initialization instead of FlexiCube, as the latter is limited by its resolution"
>
> We would like to provide further clarification as follows, to address the reviewer's concerns:
>
> 1. **The reviewer kindly suggests focusing more on reflective scenes.** Thanks for the suggestions. We have already revised our paper, restating our contribution (*L98-L107, Sec. 1*) and providing experimental results on reflective scenes (*L324-L377, Sec. 4.1*).
>
> 2. **The reviewer may think GeoSplatting's normal quality is less competitive for datasets like TensoIR and Synthetic4Relight.** We would like to highlight our performance in normal MAE is already the best for the TensoIR dataset (*L1521-L1540, Table 12 in Appendix H.4*). As for the Synthetic4Relight dataset, since this dataset does not provide normal, we cannot compare normal quality assessments on this dataset. As for visibility, we fully agree with the reviewer that it is important. We model it with occlusion map. While raytracing or neural visibility modeling would be a better solution, they typically take much longer time. Instead, Geosplatting applies occlusion map and achieves the best efficiency compared to all the inverse-rendering methods. As for the relighting performance, we want to emphasize that we have achieved the best relighting performance in both the TensoIR and Synthetic4Relight datasets (*Table 2 & Table 12*).
>
> 3. **The reviewer may question the effectiveness of the differentiable isosurface extraction method in our first stage.** We thank the reviewer for pointing this out. We have already conducted a quantitative comparison to demonstrate our method's effectiveness in enhancing geometry:
>
>    | Method | Normal MAE (TensoIR dataset) | Normal MAE (Shiny Blender dataset) |
>    | ------ | ---------------------------- | ---------------------------------- |
>    | 3DGS\* | ~10                          | ~15                                |
>    | 2DGS   | 4.17                         | 13.65                              |
>    | Ours   | **3.87**                     | **2.28**                           |
>
>    The vanilla 3DGS/2DGS optimization alone leads to suboptimal geometry, validating that our first stage is crucial, as it brings accurate geometry and precise normals that are critical for inverse rendering tasks. It is further demonstrated that our method achieves the best inverse rendering performance on reflective datasets, primarily due to the best normal brought from the first stage.
>
> 4. **The reviewer may think the 3DGS-DR method achieves better performance for reflective objects.** We thank the reviewer for bringing this related work but want to point out that 3DGS-DR is not an inverse rendering method. Nevertheless, GeoSplatting still outperforms 3DGS-DR in both rendering quality and normal estimation accuracy. We will add it to the related work.
>
>    | Method (on Shiny Blender dataset) | Normal MAE | NVS PSNR  |
>    | --------------------------------- | ---------- | --------- |
>    | 3DGS-DR                           | 4.87       | 34.09     |
>    | Ours                              | **2.28**   | **35.05** |
>
> 5. **The reviewer may think GeoSplatting's geometry quality is limited by resolution.** We acknowledge the resolution limitation but would like to highlight that even so, we still achieve the best normal quality among all the baselines.
>
> ## **Summary**
>
> We sincerely hope our explanation addresses the reviewer's concerns and will incorporate the above discussion into our revised paper. We would be happy to provide additional clarification if the reviewer has any further questions. Once again, we deeply thank you for the reviewer's hard work and would appreciate it if the reviewer could kindly reconsider our contribution.

---

### Official Review · Reviewer_T5mu · 2024-11-02

**Soundness:** 3
**Presentation:** 3
**Contribution:** 3
**Rating:** 6
**Confidence:** 4

**Summary:**

This paper proposed a novel 3DGS based inverse rendering framework, which rely on mesh-based representation as a geometry guidance. The starting point of this paper is that 3DGS cannot reconstruct accurate geometry, which is crucial for physically-based rendering. On the technical level, the main contribution of this paper is a new 3D Gaussian construction method, which aligns the gaussian primitives with the mesh by sampling 3D Gaussians on the vertices and triangle faces of the mesh.

**Strengths:**

This paper focuses on how to improve the quality of geometric reconstruction of 3DGS so that it can better meet the needs of physically-based rendering. The proposed novel isosurface guided gaussian splatting pipeline can accurately reconstruct the surface and normals of objects, thereby achieving high-quality inverse rendering results with satisfactory training time. In addition, even if the PBR part of the method is removed, the MGadapter proposed in the paper can also be considered as a new 3DGS construction method that can improve the geometric representation ability of 3DGS.

**Weaknesses:**

- The proposed method uses flexicubes as geometry guidance and requires object mask loss as supervision during training, which limits the method’s to the object level.
- For indirect lighting, the author only considered using MLP to estimate inter-reflection, but did not consider ambient occlusion.
- The quality of roughness reconstruction is not very satisfactory. This may be because the method does not take occlusion into account, so the material cannot be accurately estimated for the shadowed parts, which is also confirmed by the visualization results in the paper.
- Language issues. There are many spelling and grammatical errors in the article, even in the title of the article.

**Questions:**

- Please provide more details on the second stage optimization. Does the second stage optimization use the pruning and densification strategy in the vanilla 3DGS or just optimize the existing 3D Gaussian?
- Why can jointly training 3DGS and isosurface improve the quality of inverse rendering? Are there any experimental results to prove this? In addition, the CD socre of the mesh in Table.3 seems to be inferior to the result of directly extracting the mesh using flexicubes. Does this mean that joint training affects the reconstruction quality of the geometry?
- Given that indirect lighting is modeled as an attribute of each Gaussian, which represents the inter-reflection under the lighting conditions corresponding to the training stage, Is it possible to construct indirect lighting during relighting?

---

> ### Author Response · Authors · 2024-11-20
>
> We thank the reviewer for the recognition of our method as an innovative technique for geometry enhancement. We also greatly appreciate the reviewer’s detailed suggestions. Below, we address all the concerns point by point.
>
> ## 1. Object-Level Reconstruction vs. Scene-Level Reconstruction
>
> The reviewer is constructively concerned that employing the FlexiCubes and the object mask loss may limit the method to object-level tasks. We acknowledge this and provide a detailed analysis of the impact of object mask in Appendix F.1. Specifically, GeoSplatting utilizes FlexiCubes to enable precise geometry and normals, while also inheriting the limitation from FlexiCubes that can be applied to the object level for its current version.
>
> We agree that further extension to scene level is important and propose two promising directions. **(1)** For the object mask loss contributed to enhancing geometric stability in the optimization process (evidenced in Appendix F.1), a normal regularization technique, such as depth-normal consistency in 2DGS, can be a better alternative for scene-level extensions. **(2)** For the limited grid resolution in FlexiCubes, implementing FlexiCubes with adaptive resolution and incorporating scene contraction techniques, as seen in MipNeRF360, could be effective.
>
> ## 2 & 3. Ambient Occlusion & Roughness Reconstruction
>
> We greatly thank the reviewer for the valuable concern about GeoSplatting's challenges in handling ambient occlusion. However, as pointed out in the general response, reflective surface modeling, shadows (which also includes inter-reflection and ambient occlusion), and efficiency can be the **impossible trinity**.
>
> While we acknowledge that the quality of roughness reconstruction of GeoSplatting can be significantly affected by occlusion, Geosplatting manages to **model shadow effects by introducing a learnable occlusion term (Appendix D.2) in the PBR pipeline.** The qualitative results show that the occlusion term brings significant improvements in material-shadow decoupling, compared to those in the previously submitted version, as shown in Fig. 28-29 of the revised submission (e.g., *Hotdog, Chair* from Synthetic4Relight), with quantitative evidence provided in Table 2 (Sec. 4.2). More discussions are provided in Appendix D.2.
>
> Among the compared split-sum approaches (e.g., NVdiffrec, GS-Shader), GeoSplatting achieves the best albedo & roughness reconstruction. Moreover, **GeoSplatting excels in reflective surface modeling, achieving excellent roughness quality where all baselines fail.** This strength is evidenced in Fig. 31 (Appendix H.3), where GeoSplatting significantly outperforms other methods.
>
> Additionally, while GeoSplatting's split-sum approach struggles with roughness-shadow decoupling (Fig. 7 and Fig. 27-29), SG-based methods like TensoIR and R3DG also face significant limitations in handling specular surfaces, as illustrated in Fig. 16 (Appendix D) and Fig. 31 (Appendix H.3). We provide more discussions in Appendix D.1.
>
> ## 4. Language
>
> Thanks for the reviewer’s kind suggestion. We’ve checked the grammar and fixed a lot in our revised submission.

---

> ### Author Response · Authors · 2024-11-20
> **(Continued)**
>
> ## Q1. Second Stage
>
> The second stage of GeoSplatting is similar to the training stage of vanilla 3DGS but without any densification or pruning. Moreover, we also optimize the normals. Though it is totally free, in stage 1 it provides good initialization and in stage 2 the meaningful normal is still preserved.
>
> ## Q2.1 Why can jointly training 3DGS and isosurface improve the quality of inverse rendering?
>
> + **3DGS.** For an intuitive understanding, vanilla 3DGS is highly effective in capturing multi-view-consistent appearances of 3D objects, **not relying on normal information from light-surface interaction to guide the reconstruction.** As evidenced by Fig. 30 (Appendix H.3), when 3DGS-based methods like R3DG fail on highly specular cases, they still obtain a coarse geometry and produce a reasonable appearance.
>
> + **NVdiffrec.** In contrast, isosurface-based methods like NVdiffrec rely heavily on mesh-based differentiable PBR rendering processes, which **heavily depend on normal information from light-surface interaction to guide the reconstruction.** While this empowers NVdiffrec to reconstruct detailed lighting efficiently, the deterministic nature of isosurfaces makes it difficult to capture high-frequency light-material interactions. As a result, NVdiffrec’s failure cases on specular objects are frequently accompanied by completely noisy geometry, as illustrated in Fig. 30 (Appendix H.3).
>
> + **GeoSplatting.** Returning to our GeoSplatting approach, the Gaussian points and the mesh mutually benefit the reconstruction process. Gaussian splatting provides strong gradient properties for capturing multi-view-consistent appearances, while the mesh offers guidance for more accurate light-surface interaction estimation. A supporting example is that GaussianShader also employs a split-sum approximation, which excels in capturing environmental lighting. However, our method achieves significantly better lighting reconstruction quality, likely due to improved normal estimation, as illustrated in Fig. 31 (Appendix H.3).
>
> ## Q2.2: Does the CD scores in Table 3 mean that joint training affects the reconstruction quality of the geometry?
>
> The CD scores do not indicate a geometry quality degradation from the joint training, and we would like to clarify that the CD scores in Table 3 **have been scaled by 1e-4**. We sincerely appreciate the reviewer's careful review and apologize for the lack of clarity in the original submission.
>
> ## Q3. Inter-Reflection during Relighting
>
> Currently, indirect lighting cannot be constructed during relighting in GeoSplatting, as our proposed residual terms are highly related to the current illumination and become invalid under a different illumination.
>
> However, we believe by performing ray tracing for Monte Carlo sampling under the changed environmental lighting, the inter-reflection effects can be baked into our residual terms. While this simple technique could help improve relighting performance, we do not employ it primarily due to the inefficiency of such an extensive Monte Carlo sampling, but it raises a promising future direction and we will discuss it in the revision. Thanks for the reviewer's valuable suggestions.

---

> > ### Comment · Reviewer_T5mu · 2024-11-25
> > **Official Comment by Reviewer T5mu**
> >
> > I appreciate the authors for providing such a detailed explanation in response to my questions. And the authors' response have already address all of my concerns. Thanks again to the authors for their efforts during the rebuttal period.

---

> > > ### Author Response · Authors · 2024-11-25
> > >
> > > Thanks for all the valuable suggestions. We also greatly appreciate the reviewer's time during the rebuttal period.

---

> ### Author Response · Authors · 2024-11-28
>
> ### **Dear Reviewer T5mu,**
>
> We greatly thank for your valuable insights and helpful feedback on our work. We would like to provide a brief new quantitative comparison here, where our GeoSplating method achieves the best PSNR and normal quality (MAE) among all NeRF-based and 3DGS-based inverse rendering methods, demonstrating the effectiveness of our MGadapter as geometry guidance and highlighting our GeoSplatting's superior performance.
>
> ---
>
> | Method    | Training Time | Normal MAE (Diffuse Cases) | PSNR (Diffuse Cases) | Normal MAE (Specular Cases) | PSNR (Specular Cases) |
> | --------- | ------------- | -------------------------- | -------------------- | --------------------------- | --------------------- |
> | NeRO      | 600min        | 4.97                       | 29.08                | 2.36                        | 29.84                 |
> | TensoIR   | 270min        | 4.10                       | 34.84                | 4.42                        | 27.98                 |
> | NVdiffrec | 72min         | 4.97                       | 32.21                | 9.38                        | 28.14                 |
> | R3DG      | 140min        | 5.48                       | 33.84                | 5.48                        | 28.83                 |
> | GS-Shader | 63min         | 6.53                       | 37.54                | 7.03                        | 30.64                 |
> | Ours      | **20min**     | **3.87**                   | **38.57**            | **2.28**                    | **35.05**             |
>
> *Note that the "diffuse cases" denote scenes from the TensoIR Synthetic dataset and the "specular cases" denote scenes from the Shiny Blender dataset.*
>
> ---
>
> We sincerely thank you for recognizing and appreciating our work. We will release all the code to the community to facilitate future research. Once again, we greatly thank for your time and effort during the rebuttal phase.
>
> Best regards,
>
> Authors

---

### Official Review · Reviewer_X2HQ · 2024-11-04

**Soundness:** 2
**Presentation:** 2
**Contribution:** 2
**Rating:** 5
**Confidence:** 4

**Summary:**

This paper introduces an approach, GeoSplatting, which combines 3DGS with explicit geometric guidance and PBR. Firstly, the authors propose to extract meshes by utilizing Flexicubes. Then, the paper introduces the MGadapter, which constrains gaussians on the mesh surface. GeoSplatting incorporates a physically-based rendering framework that leverages the split-sum model to efficiently compute high-order lighting effects. This allows for fast rendering speeds and improved material and lighting disentanglement.

**Strengths:**

This paper makes full use of 3D Gaussian to optimize mesh, and propose a densification strategy. They introduce physics-based rendering equation to replace the implicit SH functions representation, which has good performance on the inverse rendering task.

**Weaknesses:**

1. The main contribution of this paper is the proposal of a hybrid representation combining mesh and Gaussian, but the discussion on mesh-based methods is not sufficiently thorough. Some omitted references are listed below, some of which are highly relevant. The authors need to state the main differences and comparisons.
   - DreamMesh4D: Video-to-4D Generation with Sparse-Controlled Gaussian-Mesh Hybrid Representation
   - Mesh-based Gaussian Splatting for Real-time Large-scale Deformation
   - High-Quality Surface Reconstruction using Gaussian Surfels
   - Direct Learning of Mesh and Appearance via 3D Gaussian Splatting
2. The comparison in terms of geometry is insufficient. The authors claim that they can reconstruct "precise" normals, which is inappropriate. Besides, here is insufficient experimental evidence to support this claim. The comparative experiments in Table 3 are too few, and the method proposed in this paper performs well only on the spot model. Including an evaluation metric for normal consistency might be better.
3. The impact of resolution on both the results and speed in the Flexicube method has not been discussed.
4. The citation formats for references within the text are improper, which causes reading difficulties.

**Questions:**

1. I have some doubts about the efficiency, as compared to 3DGS, this method requires using an MLP to learn and also needs to optimize the SDF field to extract the mesh. Could the authors please explain the reasons for the improvement in time efficiency?
2. For Eq(1), What does a heuristic function mean? So, when $\mu$ is determined, are $\mathbf{S}$ and
$\mathbf{R}$ essentially fixed?
3. The authors' experiments indicate that stage 2 is crucial for improving NVS performance, but this part does not show a significant difference compared to 3DGS.

---

> ### Author Response · Authors · 2024-11-20
>
> We thank the reviewer for the valuable suggestions. We address the concerns point by point in the responses below.
>
> ## 1. Suggested References
>
> We thank the reviews for pointing out the missing reference works. Below we provide a brief discussion about them and will add them in the revision.
>
> [1] and [2] both introduce a Mesh-to-Gaussian conversion module. However, compared to our MGadapter, their modules are **non-differentiable** and mainly aim to enable deformation through mesh-Gaussian alignment. In contrast, our pipeline is fully differentiable and mainly focuses on providing geometry guidance to enhance inverse rendering. [3] is similar to 2DGS which focuses on **geometry reconstruction tasks**, while ours targets inverse rendering challenges.
>
> [4] also includes a Mesh-to-Gaussian module like ours. However, its basic Mesh-to-Gaussian module does not focus on geometry quality and ignores two important geometric issues: **(1)** The shape of the Gaussians does not align well with the triangle mesh in tangential directions, particularly near edges. **(2)** Strict alignment in normal directions reduces appearance quality (see discussion in Appendix B.4, L798-859). In contrast, our approach overcomes both issues with an efficient and effective Mesh-to-Gaussian module.
>
> ## 2. Insufficient Geometric Comparison
>
> We appreciate the reviewer’s valuable suggestion and conduct additional experiments to support our claim (Line 084) about accurate normal estimation, where GeoSplatting achieves the best normal quality over baselines:
>
> 1. **On TensoIR Dataset**: Our method achieves the best average normal quality (mean angle error, MAE) among relightable baselines (see Table 12 in Appendix H.4, Line 1521-1540).
> 2. **On Shiny Blender Dataset**: Our approach significantly outperforms all listed inverse rendering methods in MAE (see Table 11 in Appendix H.3, Line 1426-1435).
>
> ## 3. Resolution Impact Survey
>
> Thanks for the reviewer's suggestion. We conduct comparison experiments to analyze the impact of FlexiCubes resolution on both the results and speed, which is worth discussing.
>
> Specifically, we conduct experiments on the Lego dataset using three different grid resolutions: 64, 96, and 128. We observe that when resolution increases, the normal MAE as well as NVS PSNR improves a little (~0.5 in MAE and ~0.4 in NVS). However, the training time increases (e.g., by a factor of 4x from 96 to 128), which may not justify the potential benefits. Therefore, it indicates the trade-off between performance vs. resolution, which justifies the choice of grid size as 96. We provide details results and discussions in Appendix E.
>
> ## 4. Writing Issues
>
> We thank the reviewer for the kind suggestion about the incorrect citation format and have fixed it in our revised submission.

---

> ### Author Response · Authors · 2024-11-20
> **(Continued)**
>
> ## Q1. Efficiency
>
> The reviewer valuably questions the efficiency of our method since our training time is 20 minutes, compared to 15 minutes for vanilla 3DGS with the same CUDA implementation, which we explain as follows:
>
> - **We use the HashGrid from Instant-NGP (implemented with tiny-cuda-nn) instead of a simple MLP**, enabling very fast attribute querying. Details of the network configuration are provided in Appendix B.5.
> - **We employ FlexiCubes (scalar field) instead of an SDF field**. **(1)** Unlike the SDF field, the scalar field does not require maintaining SDF properties (e.g., no need for an eikonal loss), making optimization more efficient. **(2)** Moreover, unlike neural field methods like NeuS that represent the SDF field by an MLP, our scalar field uses a 96x96x96 grid with iso-values stored at grid vertices. Directly optimizing these iso-values is much faster than training an SDF MLP.
>
> ## Q2. Heuristic Function
>
> We apologize for the lack of clarity in the submitted version and make a detailed explanation as follows:
>
> - Specifically, for a given value of $\mu$, a set of barycentric factors is uniquely determined.
> - From these factors, $S$ and $R$ are computed based on the shape of each triangle.
> - A more detailed formulation can be found in Appendix B.2.
>
> ## Q3. Second Stage
>
> The second stage is similar to the training process of vanilla 3DGS, with key differences being that: **(1)** No densification or pruning is involved. **(2)** Gaussian points with normal attributes are optimized under PBR constraints, leading to limited movement.
>
> While vanilla 3DGS’s free optimization enhances NVS quality, the light-material-interaction-aware geometry guidance in GeoSplatting’s first stage is essential for handling non-Lambertian surfaces. As evidenced by Table 10 (Appendix H.3), Vanilla 3DGS, with its free optimization and densification strategy, struggles to estimate glossy surfaces, whereas GeoSplatting’s geometry guidance can avoid these issues.
>
>
>
> [1] *Li, et al. "DreamMesh4D: Video-to-4D Generation with Sparse-Controlled Gaussian-Mesh Hybrid Representation." arXiv (2024).*
>
> [2] *Gao, Lin, et al. "Mesh-based Gaussian Splatting for Real-time Large-scale Deformation." arXiv (2024).*
>
> [3] *Dai, Pinxuan, et al. "High-quality Surface Reconstruction using Gaussian Surfels." SIGGRAPH (2024).*
>
> [4] *Lin, et al. "Direct Learning of Mesh and Appearance via 3D Gaussian Splatting." arXiv (2024).*

---

> > ### Author Response · Authors · 2024-11-25
> >
> > Dear Reviewer X2HQ,
> >
> > We thank a lot for the reviewer's time during the rebuttal. We sincerely hope that the reviewer's concerns about missing references, geometry quality, resolution impact, and writing issues are well addressed, and we are willing to provide detailed explanations if there are any further concerns raised by the reviewer. We would appreciate it if the reviewer could provide more feedback and kindly reconsider the contribution of our work. Thank you again!
> >
> > Best wishes,
> >
> > Authors

---

> ### Author Response · Authors · 2024-11-28
> **Request for Further Feedback during Extended Rebuttal Phase**
>
> ### **Dear Reviewer X2HQ,**
>
> We sincerely thank you for your time and effort during the rebuttal period and would greatly appreciate it if you could provide feedback on our responses. We would also welcome the opportunity to discuss any remaining concerns.
>
> We would like to provide a brief quantitative comparison here, to address your earlier concern regarding potentially inadequent experimental results. We would like to highlight that our GeoSplating method achieves the best PSNR and normal quality (MAE) among all NeRF-based and 3DGS-based inverse rendering methods, demonstrating our approach as a state-of-the-art inverse rendering method for reflective scenes and validating GeoSplatting's excellent ability in capturing high-frequency lighting-material interactions.
>
> ---
>
> | Method    | Training Time | Normal MAE (Diffuse Cases) | PSNR (Diffuse Cases) | Normal MAE (Specular Cases) | PSNR (Specular Cases) |
> | --------- | ------------- | -------------------------- | -------------------- | --------------------------- | --------------------- |
> | NeRO      | 600min        | 4.97                       | 29.08                | 2.36                        | 29.84                 |
> | TensoIR   | 270min        | 4.10                       | 34.84                | 4.42                        | 27.98                 |
> | NVdiffrec | 72min         | 4.97                       | 32.21                | 9.38                        | 28.14                 |
> | R3DG      | 140min        | 5.48                       | 33.84                | 5.48                        | 28.83                 |
> | GS-Shader | 63min         | 6.53                       | 37.54                | 7.03                        | 30.64                 |
> | Ours      | **20min**     | **3.87**                   | **38.57**            | **2.28**                    | **35.05**             |
>
> *Note that the "diffuse cases" denote scenes from the TensoIR Synthetic dataset and the "specular cases" denote scenes from the Shiny Blender dataset.*
>
> ---
>
> In light of these results, we sincerely hope that you could reconsider our contributions as we believe that GeoSplatting proposes a novel framwork, employing geometry guidance to both efficienctly and effectively enhance inverse rendering ability, which we see valuable to the community. We will also release all the code to facilitate future research. We are more than happy to address any additional feedback you may have.
>
> Best regards,
>
> Authors

---

> > ### Comment · Reviewer_X2HQ · 2024-11-30
> > **Response to Authors**
> >
> > Dear AC and authors,
> >
> > I greatly appreciate the authors' detailed response and the explanation of the related experiments.
> > My main concern remains that the proposed method reconstructs geometry by combining Instant-NGP and FlexiCubes, while restricting GS to a triangular mesh. The authors' experiments also show that this approach results in worse rendering performance compared to the original 3DGS method. A second stage using the original 3DGS to enhance the rendering effect is still required. Therefore, I continue to believe that geometry and rendering are separate processes, and I remain skeptical of the contribution. As a result, I am maintaining my original score.
> >
> > Thanks for your hard work.

---

> > > ### Author Response · Authors · 2024-11-30
> > >
> > > ## **Dear Reviewer X2HQ,**
> > >
> > > We are happy most concerns of the reviewer have been addressed and sincerely thank the reviewer for the time and effort in evaluating our work.
> > >
> > > The reviewer kindly provided new feedback and raised concerns that:
> > >
> > > 1. "... this approach results in worse rendering performance compared to the original 3DGS method"
> > > 2. "... the geometry and rendering are separate processes", leading to "skeptical of the contribution".
> > >
> > > We summarize the reviewer's concern as follows:
> > >
> > > 1. The reviewer may think our inferior rendering performance relative to vanilla 3DGS diminishes our contribution, as the reviewer is concerned our improved final performance is primarily due to the second stage rather than our proposed first stage.
> > > 2. The reviewer may think our isosurface geometry reconstruction in stage 1 and 3DGS rendering in stage 2 are separate processes, leading to concerns about the effectiveness of our first stage.
> > >
> > > We appreciate the reviewer's insightful concerns and would like to provide a more comprehensive explanation for clarification, hoping to seek common ground with the reviewer:
> > >
> > > ---
> > >
> > > ## **1. Inferior rendering performance relative to vanilla 3DGS does not diminish our contribution, because:**
> > >
> > > **The goal of our work focuses on inverse rendering (IR) tasks instead of novel view synthesis (NVS). While IR also enables the rendering from new views, it differs from NVS because IR involves decomposing geometry, spatially varying materials, and lighting from images, which introduces a tradeoff between decomposition performance and NVS quality.** As evidence, no existing NeRF-based or 3DGS-based inverse rendering method can achieve superior rendering quality compared to vanilla 3DGS (designed only for NVS) across general cases from the NeRF Synthetic or TensoIR Synthetic datasets.
> > >
> > > **We already achieved superior rendering quality compared to all 3DGS-based inverse rendering methods, as shown in Table 1 and Table 12.** Moreover, we would like to emphasize that the improved final performance is not primarily due to the second stage. Instead, the proposed first stage plays a vital performance, as the isosurface guidance introduced in Stage 1 brings precise normal estimation which significantly enhances the 3DGS framework's inverse rendering performance. If the second stage (free 3DGS optimization) is indeed the primary contributor, it would be contradictory that no other 3DGS-based inverse rendering method (e.g., R3GD, GaussianShader) manages to surpass GeoSplatting in rendering quality.
> > >
> > > ---
> > >
> > > ## **2. Isosurface geometry and 3DGS rendering are mutually enhanced, because:**
> > >
> > > **While vanilla 3DGS/2DGS optimization alone leads to suboptimal geometry, the proposed first stage is crucial, as it brings accurate geometry and precise normals that are critical for inverse rendering tasks.** Extensive results demonstrate our method's effectiveness in enhancing geometry. For a more comprehensive comparison, we have included results for 2DGS as well.
> > >
> > > | Method | Normal MAE (Diffuse Cases) | Normal MAE (Specular Cases) |
> > > | ------ | -------------------------- | --------------------------- |
> > > | 3DGS\* | ~10                        | ~15                         |
> > > | 2DGS   | 4.17                       | 13.65                       |
> > > | Ours   | **3.87**                   | **2.28**                    |
> > >
> > > *\*For 3DGS, we could only obtain pseudo normals from the depth map.*
> > >
> > > **Our geometry and rendering are mutually enhanced in inverse rendering tasks even at the first isosurface stage, as demonstrated by superior rendering quality and geometry accuracy compared to 3DGS/2DGS in reflective cases.** With superior rendering quality and geometry accuracy in Stage 1, we would respectfully argue that our geometry and rendering are not separate processes. Quantitative results on specular cases further support our claim of mutual enhancement between geometry and rendering in our first stage.
> > >
> > > | Method              | PSNR (Specular Cases) | Normal MAE (Specular Cases) |
> > > | ------------------- | --------------------- | --------------------------- |
> > > | 3DGS\*              | 30.37                 | ~15                         |
> > > | 2DGS                | 29.47                 | 13.65                       |
> > > | Ours (only stage 1) | **32.09**             | **4.04**                    |
> > >
> > > *\*For 3DGS, we could only obtain pseudo normals from the depth map.*
> > >
> > > **As a result, we believe our pipeline, though containing 2 stages, provides a deliberated design that both geometry guidance and 3DGS rendering enhance each other.**
> > >
> > >
> > > ## **Summary**
> > >
> > > We sincerely hope our explanation addresses the reviewer’s concerns and will incorporate the above discussion into our revised paper. We would be happy to provide additional clarification if the reviewer has any further questions. Once again, we deeply thank you for the reviewer’s hard work and would appreciate it if the reviewer could kindly reconsider our contribution.

---

> > > ### Author Response · Authors · 2024-12-02
> > >
> > > Dear Reviewer X2HQ,
> > >
> > > We sincerely thank you for your valuable feedback, time, and effort during the rebuttal period. We greatly appreciate the insightful discussion, which has significantly strengthened our paper. Following your suggestions, we have added experiments to demonstrate the superior geometry of our method. Additionally, we have included further explanations to show how geometry and rendering enhance each other. We hope our responses have fully addressed your concerns. If there are any remaining issues, we are happy to provide further clarifications or additional experiments as needed. We would greatly appreciate it if you could consider raising the rating of our paper.
> > >
> > > Thank you!
> > >
> > > Best regards,
> > > The Authors

---

### Author Response · Authors · 2024-11-20
**General Response**

We sincerely thank all the reviewers for their valuable and constructive feedback. Below we provide a general response to address shared concerns. We also provide point-to-point responses to each reviewer, as well as a revised submission that contains additional experimental results and detailed discussions.


# Shadow Modeling (Reviewer T5mu, f2Bn)

The reviewers valuably raised concerns about GeoSplatting’s limitations in shadow modeling and worried that it might not reflect the state-of-the-art performance in inverse rendering. However, we respectfully argue that in the inverse rendering field, there exists an **“impossible trinity”**, where, to the best of our knowledge, none of the existing works can address **reflective surface, shadow, and efficiency** at the same time.

Specifically, existing inverse rendering works can be categorized into 3 groups based on their inverse rendering equations Monte Carlo (MC) sampling, split-sum approximation, and Spherical Gaussian (SG). Among these methods,

1. **MC-based methods** (e.g., NVdiffrecmc) can model both shadow and reflective surfaces but suffer from a limited number of samples due to high computational costs, leading to efficiency issues.
2. **Split-sum-based methods** (e.g., NVdiffrec, GS-Shader) efficiently handle complex lighting but struggle with shadows, as they assume minimal self-occlusion and ignore the visibility term in integration.
3. **Spherical Gaussian-based methods** (e.g., TensoIR, R3DG) handle inter-reflections and ambient occlusion well but have difficulty reconstructing highly reflective surfaces due to the limitations of SG in capturing high-frequency lighting interactions.

**Our GeoSplatting applies a split-sum model. While we acknowledge its limitation in shadow modeling, we show that:**

1. GeoSplatting achieves state-of-the-art performance with excellent ability in reflective surface modeling and efficiency, as evidenced by experimental results in Appendix H.3 (Line 1413-1511).

2. Further, following NVdiffrec, we also introduce occlusion terms in our PBR pipeline to mitigate the issue and achieve comparable decomposition performance while keeping the best relighting performance, as shown in Fig. 32 (Appendix H.4, Line 1541-1563).
3. Moreover, it also achieves the best normal performance (Table 12 in Appendix H.4, Line 1521-1540).

**We would see these contributions as valuable to the community** and more details on split-sum vs. SG methods are discussed in Appendix D.


# Geometry Quality (Reviewer X2HQ, DXwn)

The reviewers constructively raised concerns about the geometry quality of GeoSplatting. While Table 3 only demonstrates comparable geometry recovery performance in terms of CD and F-score, in the context of inverse rendering, the normal quality could be more crucial. Therefore, we provide additional experimental results on two datasets, which both indicate **our superior normal performance**.

1. **On TensoIR Dataset**: Our method achieves the best average normal quality (mean angle error, MAE) among relightable baselines (see Table 12 in Appendix H.4, Line 1521-1540).

2. **On Shiny Blender Dataset**: Our approach significantly outperforms all listed inverse rendering methods in MAE (see Table 11 in Appendix H.3, Line 1426-1435).

---

> ### Comment · Reviewer_f2Bn · 2024-11-23
>
> I appreciate the authors' explanation of shadow modeling and agree with some of their statements. However, I respectfully disagree with certain points.
>
> Firstly, while I commend the authors for correctly categorizing current inverse rendering methods, I believe that the three approximations to the rendering equation they discuss do not fundamentally lead to significant differences. For instance, NeRO uses a split-sum approach in the first stage to obtain a geometry prior for reflective objects. Yet, the actual contribution to geometry comes from the combination of NeuS and Ref-NeRF's IDE encoding. In the second stage, NeRO employs MC but does not address the issue of shadow baking in the albedo.
>
> Furthermore, I think the representative method for the split-sum approach is Neural-PIL and NeRO; and for SG, it is PhySG and InvRender. Their ability to model indirect illumination is not due to SG themselves but to the methods proposed by InvRender. Moreover, SG-based methods cannot achieve shadow removal; it was the subsequent method, RobIR, that achieved high-quality visibility modeling.
>
> Regarding Geo-Splatting, the authors emphasized during the rebuttal phase that their method performs well not only in general scenes but also has significant advantages in reflective scenes. Based on this premise, I see issues with the contributions they claim.
>
> - GeoSplatting is not the state-of-the-art for reflective scenes. Methods like NeRF-Casting, UniSDF, ENVIDR, GIR (https://arxiv.org/abs/2312.05133) and even GS-DR (based on 3D-GS) achieve better results than GeoSplatting in both visual quality and quantitative metrics. I am more inclined to agree that Geo-Splatting is the SOTA for reflective scenes based on inverse rendering. Additionally, in Table 11 and Figure 30, Geo-Splatting exhibits excessive smoothing in the `helmet` scene, making it inferior to NeRO (it could be better to add the geometry of NeRO in the corresponding figures and tables).
>
> - While I agree with the authors' perspective if we consider only quantitative metrics, inverse rendering is not a field where a single metric can explain everything. In Figure 32, there is a noticeable hue inconsistency in relighting, which makes it difficult to consider this a better result than other methods. Furthermore, Figure 32 does not include a comparison with TensoIR.

---

> ### Author Response · Authors · 2024-11-23
> **Reply to Reviewer f2Bn**
>
> We sincerely thank Reviewer f2Bn for the insightful discussion on shadow modeling. The reviewer mentioned many approaches that make a great effort in shadow modeling. To further address the reviewer's concern, we would like to rephrase and clarify some potentially misleading points in our previous explanations.
>
> ------
>
> ### Discussion: Relationship between PBR Approximations with Shadow Modeling
>
> As our previous explanation may have been overly brief and therefore misleading, we first clarify the relationship between the different approximations and the challenges in shadow modeling.
>
> While highly appreciating the great effort made by the existing works (e.g., Neural-PIL, NeRO, PhySG, InvRender), we also strongly agree with Reviewer f2Bn that, it is not the approximations themselves that fundamentally *enable* certain outcomes. Rather, we believe it is the approximations that fundamentally *limit* what can be achieved.
>
> For instance, as the reviewer noted, one of the most prominent SG-based methods, *InvRender*, introduces a crucial component to model visibility (Sec. 3.2 in *InvRender*). In our opinion, the same technique—or even methodology—cannot be straightforwardly adapted to split-sum-based methods. This limitation arises primarily from the non-decomposability inherent in split-sum-based lighting. We refer to this challenge as our **“impossible trinity”**, highlighting how split-sum-based methods face significant difficulties in modeling visibility effectively.
>
> When discussing shadow modeling, the fundamental distinction between split-sum and SG lies in the decomposability of light. In SG approximations, the lighting representation is decomposable (as shown in Eq. 2 of *InvRender*), making it possible to perform an approximation that leads to Eq. 3 in *InvRender*.
>
> $$ L_\text{SG}(\boldsymbol{\omega_i})=\sum_{k=1}^SG(\boldsymbol{\omega_i};\boldsymbol{\xi_k},\lambda_k,\boldsymbol{\mu_k})\quad\text{(w/o visibility term)} $$
>
> $$ L_\text{SG}(\boldsymbol{x_i},\boldsymbol{\omega_i})=V(\boldsymbol{x_i},\boldsymbol{\omega_i})\sum_{k=1}^SG(\boldsymbol{\omega_i};\boldsymbol{\xi_k},\lambda_k,\boldsymbol{\mu_k})\quad\text{(w/ visibility term)} $$
>
> In contrast, the split-sum approximation is non-decomposable (as indicated by Eq. 7 in *NeRO*). This non-decomposability means that exchanging the order of the visibility term and the integral introduces significant errors.
>
> $$ L_\text{split-sum}(\boldsymbol{\omega_i})=\int_{\Omega(\boldsymbol{\omega_i})} I_\text{cubemap}(\boldsymbol{\omega})D_\text{GGX}(\rho,\text{reflect}(\boldsymbol{\omega}))\text{d}\boldsymbol{\omega} \quad\text{(w/o visibility term)} $$
>
> $$ \begin{align*}L_\text{split-sum}(\boldsymbol{x},\boldsymbol{\omega_i})&=\int_{\Omega(\boldsymbol{\omega_i})} V(\boldsymbol{x},\boldsymbol{\omega})I_\text{cubemap}(\boldsymbol{\omega})D_\text{GGX}(\rho,\text{reflect}(\boldsymbol{\omega}))\text{d}\boldsymbol{\omega} \\\\&\ne V(\boldsymbol{x},\boldsymbol{\omega_i})\int_{\Omega(\boldsymbol{\omega_i})} I_\text{cubemap}(\boldsymbol{\omega})D_\text{GGX}(\rho,\text{reflect}(\boldsymbol{\omega})))\text{d}\boldsymbol{\omega} \quad\text{(w/ visibility term)}\end{align*} $$
>
> Although *NeRO*, which employs a neural light integral approximation similar to split-sum (Eq. 10 in *NeRO*), uses Monte Carlo (MC) sampling in its second stage, its shadow modeling capabilities are still inferior to those of state-of-the-art SG-based methods, *RobIR*, as the reviewer pointed out. We believe this is likely due to the inherent limitations of split-sum. Eq. 10 will introduce noticeable errors for high-frequency lighting, making it impossible to adopt similar visibility modeling techniques from SG-based methods to split-sum-based NeRO. Similarly, while *RobIR* demonstrates impressive shadow modeling performance, the reconstructed environment map remains highly blurred (Fig. 10), which stems from the low-frequency nature of SG and represents SG's fundamental limitation.
>
> While we would emphasize the challenges of shadow modeling in our split-sum-based GeoSplatting by analyzing the “impossible trinity”, we also want to highlight the significant contributions and inspiring advancements made by NeRO and RobIR. Despite the limitations of split-sum, the techniques employed in NeRO offer promising directions for modeling visibility by utilizing different approximations across multiple stages, highlighting a potential avenue for future exploration in our work.

---

> ### Author Response · Authors · 2024-11-23
> **(Continued) Reply to Reviewer f2Bn**
>
> ### Rebuttal: Contribution Claims
>
> We sincerely thank the reviewer for the thoughtful suggestion and would like to make a fair and precise claim as follows:
>
> GeoSplatting is a novel inverse rendering method that leverages a hybrid representation of 3DGS and isosurfaces. By inheriting the excellent efficiency of 3DGS and incorporating accurate isosurface geometry guidance, GeoSplatting achieves state-of-the-art inverse rendering performance in reflective scenes, which demonstrates its impressive ability in modeling high-frequency light-material interaction. Additionally, for general cases, GeoSplatting delivers decomposition results that are comparable to those of state-of-the-art inverse rendering baselines. Extensive quantitative and qualitative results are provided to support our claims.
>
> ---
>
> ### Rebuttal: NeRO
>
> As *NeRO* requires approximately 25 hours to train a single case on an RTX 2080Ti (as reported in the original paper), we will include its geometry results in Table 11 and Figure 30 in the revised version of our paper. Regarding excessive smoothing in the Helmet scene, the limited resolution may prevent GeoSplatting from reconstructing the detailed geometry near the seam.
>
> ---
>
> ### Rebuttal: Results in Figure 32
>
> We acknowledge that despite of higher PSNR, the hue inconsistency in GeoSplatting's qualitative results can be noticeable. It is still primarily due to GeoSplatting's subpar ability in shadow modeling.
>
> The *Armadillo* scene shows the strongest shadow effects among the eight scenes in the Synthetic4Relight and TensoIR datasets. The hue inconsistency in relighting, which is constructively noted by the reviewer, is due to these shadow effects. Specifically, the ground-truth environment map for the training view features a strong blue hue. Underestimating the shadow effects results in the environment map being baked into the albedo, leading to less red-toned outputs. We will include additional discussion in the revised manuscript to present this as a failure case (as further supported by the quantitative results in Table 12). Further, we add the relighting results of TensoIR to Fig. 32. As illustrated, the TensoIR achieves better decomposition under such strong shadow effects.
>
> ---
>
> ### Thanks
>
> Thanks for all the valuable suggestions and we hope the discussion can help with a more clear understanding.

---

### Author Response · Authors · 2024-11-26
**Rebuttal Summary**

### **Dear Reviewers, ACs, and PCs,**

We sincerely thank all the reviewers for their valuable and constructive feedback, as well as for dedicating their time to reviewing our paper. Based on the insightful suggestions provided during the rebuttal phase, we have conducted a thorough revision addressing the reviewers' key concerns. Below, we summarize these concerns and detail the revisions and updates included in the final submission. We hope this provides the reviewers, ACs, and PCs with a clearer understanding of the progress and outcomes of the rebuttal discussion.

---

### **Revisions**

+ **[Abstract, Introduction]** We restate our contribution to make it precise, according to Reviewer f2Bn's valuable suggestions. *[L10-L32, Abstract] [L98-L107, Section 1]*
+ **[Related Work]** We rewrite the section and reference the missing related work mentioned by Reviewer X2HQ and f2Bn. *[L122-L161, Section 2]*
+ **[Experiments]** We replace the NeRF NVS experiment with the Shiny Blender experiment in Sec. 4.1 to address Reviewer f2Bn's concern. *[L324-L377, Section 4.1]*
+ **[Experiments]** We remove the misleading Avg. column in Sec. 4.3 to address concerns raised by Reviewer X2HQ and DXwn. *[L458-L464, Section 4.3]*
+ **[Conclusion]** We restate our conclusion and de-emphasize performance in general cases, according to Reviewer f2Bn's valuable suggestions. *[L535-L539, Conclusion]*

---

### **Concerns**

+ **[X2HQ, DXwn]** Insufficient geometry comparison to claim "precise normal".
  + **[Authors]** Additional experiments support our claims. *[L367-L377, Table 1] [L1521-L1540, Table 12]*

+ **[X2HQ, f2Bn]** Missing reference.
  + **[Authors]** We rewrite the related work section and reference the related work. *[L122-L161, Section 2]*
+ **[T5mu]** Object-level limitation.
  + **[Authors]** We acknowledge and provide a detailed analysis. *[Appendix F]*
  + **[Authors]** We propose promising directions for scene-level extensions. *[Response to X2HQ]*
+ **[T5mu, f2Bn]** Poor shadow modeling.
  + **[Authors]** It is not what our work mainly focuses on. We explain it in both the General Response and the appendix. *[General Response: Shadow Modeling] [Appendix D]*
  + **[Authors]** We introduce a learnable occlusion term to mitigate it and conduct an analysis. *[Appendix D.2]*
+ **[DXwn]** Lack of novelty.
  + **[Authors]** We explain in the response. *[Response to X2HQ]*
+ **[f2Bn]** NVS comparison may not be critical for inverse rendering.
  + **[Authors]** We explain in the response. *[Response to f2Bn]*
  + **[Authors]** We replace the NeRF NVS results with Shiny Blender NVS results accordingly. *[L324-L377, Section 4.1]*

---

### **Other Points**

+ **[X2HQ]** Resolution analysis. *[Appendix E]*
+ **[X2HQ]** Incorrect citation formats. *[Corrected]*
+ **[f2Bn]** Dataset choice. *[Appendix D.2] [Appendix H.4]*
+ **[T5mu, DXwn]** Typos. *[Corrected]*
+ **[f2Bn]** Incorrect training time metrics. *[Corrected]*

---

### **Summary**

We sincerely thank the reviewers for their valuable suggestions, which have helped us strengthen our revised submission. We believe the revised submission could be much more robust and contribute to broadening the field of 3DGS-based inverse rendering, offering valuable insights to the community. We deeply appreciate the efforts of the reviewers, ACs, and PCs during the rebuttal period.

*Best regards,*

*Authors*

---

### Meta-Review · Area_Chair_krjN · 2024-12-18

**Metareview:**

The paper introduces a novel hybrid representation that combines 3DGS with isosurface-based geometry guidance and a differentiable PBR equation. The proposed approach improves performance in reflective scenes, enhance geometry quality, and maintain efficiency across diverse datasets. While all reviewers appreciated the authors' efforts and the detailed rebuttal, there are several limitations and concerns:

1. Limited Tasks: The representation utilizes FlexiCubes, which may lead to issues when handling complex scenes or objects that are difficult to represent with triangle meshes. Moreover, the use of an additional mask makes scene-level tasks difficult.

2. Suboptimal Decomposed PBR Material: Despite achieving positive final rendering results, the decomposed PBR material textures are not clean. The high-quality decomposition is crucial for realistic rendering and material representation.

3. Shadow issue: The occlusion is not well distinguished.

4. Limited Technical Contribution: The proposed method's main innovation lies in combining 3DGS with isosurface-based geometry guidance to boost performance, which may be viewed as an incremental enhancement rather than a substantial technical advancement.

Given these concerns, I do not recommend the acceptance of this paper.

**Additional Comments On Reviewer Discussion:**

Several concerns have been raised by the reviewers regarding the paper, spanning from the introduction and related work to the experiments and conclusion. Most of them have been clarified by the authors in their revision. However, there are several primary issues: the limited application cases, the unclean PBR material decomposition, the shadow issues, and the limited contribution.

During the rebuttal, the authors made considerable efforts to clarify unclear parts and demonstrated their contributions through numerous experiments. Despite these efforts, most reviewers remain concerned about the limitations mentioned above and have maintained their negative evaluations. Therefore, I do not recommend accepting this paper.

---

### Decision · Program_Chairs · 2025-01-22

Reject